# Wind Turbine Noise Behaviorally and Physiologically Changes Male Frogs

**DOI:** 10.3390/biology11040516

**Published:** 2022-03-27

**Authors:** Jun-Kyu Park, Yuno Do

**Affiliations:** Department of Biological Science, Kongju National University, Gongju 32588, Korea; pjk8578@smail.kongju.ac.kr

**Keywords:** acoustic signal, amphibians, calling pattern, endocrine disruption, immunity, wind turbine noise

## Abstract

**Simple Summary:**

We analyzed the behavioral–physiological–immunological interconnected process of Japanese tree frogs (*Dryophytes japonicus*) during their breeding season when exposed to wind turbine noise. Frogs collected from paddy fields with wind power generators exhibited a faster call rate, higher salivary concentrations of corticosterone, and lower innate immunity. However, frogs exposed to the turbine noise for a short period of time only showed faster call rates. An increase in corticosterone was correlated with an increase in call rate and a decrease in immunity. It seems that the need for energy mobilization due to an increase in the call rate leads to a decrease in innate immunity through an increase in corticosterone. The decrease in immunity due to energy tradeoff or physiological response can change disease epidemiology in the population and create new adaptive patterns within these habitats.

**Abstract:**

As the advantages of wind energy as an eco-friendly strategy for power generation continue to be revealed, the number of offshore wind farms also increases worldwide. However, wind turbines can induce behavioral and physiological responses in animals by emitting various types of noises. In this study, we investigated the behavioral, physiological, and immunological responses of male Japanese tree frogs (*Dryophytes japonicus*) when exposed to wind turbine noise. To determine the effects during the breeding season, frogs were collected from areas with and without wind turbines. Additionally, we exposed the frogs to recorded wind turbine noise at a site without a wind generator for 1 h to 24 h to analyze the short-term effects. Three types of calling patterns (dominant frequency, note duration, and call rate) were analyzed to investigate behavioral responses. Physiological responses were assessed using two steroid hormones assays, namely testosterone and corticosterone detection in the saliva. The immunity of each individual was assessed using a bacterial killing assay. The wind turbine group in the field had a higher call rate and corticosterone levels and lower immunity than the group in the field without turbines present, and all three of these variables were correlated with each other. Conversely, in the noise exposure experiment, a higher call rate was only observed post-exposure compared to pre-exposure. Thus, turbine noise seems to induce decreased immunity in Japanese tree frogs as an increase in energy investment that triggers a behavioral response rather than acting as a sole physiological response that leads to a direct increase in corticosterone. This decreased immunity due to energy tradeoff or physiological response can change the disease epidemiology of the population and create new adaptive patterns in these habitats.

## 1. Introduction

Wind energy, as a renewable and eco-friendly energy source, has garnered significant global interest due to several advantages, including relative cost competitiveness and technological maturity [1]. With the increase in global energy demand, the installation and technology of wind power generators are also improving, and wind power use is expected to continue growing in the coming years [2]. Although wind generation is a clean energy source that is known to have no emissions of greenhouse gas or hazardous waste, it has the potential to create several ecological challenges [3,4]. For instance, the construction of wind farms degrades habitat quality, and the wind turbines can collide directly with many seabird populations and bats [5,6]. Additionally, wind power generation emits various types of noise, each of which can induce different stresses [7,8]. Moreover, due to the noise generated by wind turbines, as well as the lack of physical space and efficiency of wind conditions, offshore wind farm construction is increasing [3,9]. Accordingly, the impact of wind power generation on marine ecosystems is an emerging concern, prompting studies focusing on the impact of habitat degradation, noise exposure response, and reduced reproductive capacity on marine organisms, such as marine mammals, fishes, birds, and algae [10,11,12,13]. However, development of offshore wind farms could potentially cause significant detrimental effects to freshwater animals that live near paddy fields, reservoirs, or streams housing the wind power generators. Therefore, considering the dearth of information regarding the potential impacts of wind energy development on entire ecosystems [4], studies from various ecological perspectives are necessary.

Among the various noise types associated with wind power generators, the low-frequency mechanical noise (approximately 100–500 Hz) from the nacelles and generator is generally inaudible to humans. Additionally, the blade noise generated by the rotation of the wind turbine blade is approximately 500–1000 Hz, and the aerodynamic noise generated by the interaction between the blade and wind is approximately 2000–8000 Hz [14]. Among these noise types, low-frequency noise can act as stressor that causes endocrine responses in an organism [15,16], and its adverse health effects have been verified [17]. In contrast to inaudible low-frequency noise, audible turbines and aerodynamic noise represent the main sources of noise produced by wind power generators and reportedly cause annoyance and mental stress in humans [18]. In addition to humans, these noises can potentially affect other animals that hear or interact with sounds. Graylag geese (*Anser anser*) raised near wind turbines were physiologically affected by the noise itself, so they had higher cortisol levels in blood and lower weight gain than geese raised away from turbines. [19]. Turbine noise also affects the vocal communication of the greater prairie chicken (*Tympanuchus cupido pinnatus*) by changing the soundscape of its surroundings [20]. Additionally, turbine noise actually induces increases in cortisol through the hypothalamic–pituitary–adrenal (HPA) axis response of the European badger (*Meles meles*), which the researchers suggest could alter the risk of infection or disease in the population by changing the immune system [21].

Frogs that live in various types of freshwater habitats assemble at specific locations during the breeding season and use acoustic signals to find mates [22]. Therefore, wind turbine noise during the breeding season may negatively affect the mating of frogs. If the sound pressure level of the noise during this period is large or the frequency of noise overlaps with that of the calling in a frog species, a masking effect that dilutes the acoustic signal can occur at breeding sites [23,24]. Although select studies have reported on the masking effect of noise on various biological responses in frogs, these studies have focused primarily on noise emitted by traffic [25,26,27] and airplanes [28]. These noises can show strong pulses in a short time, with irregular intervals, whereas the noise generated by wind turbines can be persistent and of relatively low intensity. Additionally, it is necessary to understand whether wind turbine noise acts as a single stressor or affects species through other complicated processes.

Male frogs can enhance advertisement calling to avoid these masking effects [25,29,30], a response that could be physiologically associated with corticosterone, a glucocorticoid hormone that aids in glucose mobilization [31,32,33]. Additionally, noise can directly affect the endocrine system of frogs [26,34]. A change in the endocrine response due to stress leads to increased corticosterone by stimulation of the HPA axis [33,35]. Whether these two effects act alone or in combination, a momentary increase in corticosterone allows the animal to respond appropriately to a threatening situation by assisting in the immediate coping mechanism caused by metabolic changes and the allocation of required energy [36]. However, the chronic increase in corticosterone due to prolonged energy tradeoffs caused by stress can lead to reduced immunity by a maladaptive response, such as imbalances in energy allocation [26,35].

In the current study, we investigated the behavioral and physiological responses of male Japanese tree frogs (*Dryophytes japonicus*) caused by wind turbine noise in paddy fields. Japanese tree frogs are a prolonged breeding species that make an advertisement call during a relatively long period in the breeding season compared to an explosive breeding species that engages in calling for a short period. Additionally, these tree frogs inhabit paddy fields only during their breeding season [37,38]. Thus, the prolonged breeding species may be more exposed to constant noise during reproduction.

We assumed that the wind turbine noise could change the pattern of frog calling. It was also expected that this would alter the physiological state for energy mobilization and lead to a decrease in immunity. Herein, to confirm the effect of wind turbine noise, the noise frequency of the wind power generator was analyzed and compared with the frequency of advertisement calling from Japanese tree frogs. Changes in calling patterns induced by the overlapping frequencies were then identified to better understand the behavioral response of frogs according to the noise of the wind turbines. Finally, we studied the response of steroid hormones and the immune system to turbine noise. The advertisement calling of frogs, saliva hormones, and serum innate immunity were analyzed to identify these responses. Field investigation and noise exposure analysis of paddy fields were conducted in parallel to identify whether the aerodynamic noise of wind turbines alone cause behavioral and physiological responses. Collectively, this study aimed to understand the process by which frogs respond and adapt to noise during the breeding season by analyzing their behavioral and physiological responses.

## 2. Materials and Methods

### 2.1. Field Investigation

In South Korea, Japanese tree frogs breed from April to the end of July [39,40]. However, we confirmed that the advertisement calling of this frog species occurred from the end of May to the middle of August in 2020 at Yeonggwang-gun, which is the wind farm site of our field investigation. Thus, the experiment was conducted in July, which was the middle to end of the breeding season in 2020. We investigated 27 male frogs in three wind farm areas (each investigation area was about 1.7 to 1.8 km^2^) with a wind power generator (2.0 MW) from Yeonggwang-gun (wind turbine group). There were eight to fifteen wind turbines in each area. Each wind turbine was arranged in a line with a distance of 220–270 m between them in the paddy fields. The wind turbines at this site operated at wind speeds of up to 2 m/s. The wind speed at the time of analysis was 2–4 m/s. All experiments were carried out immediately after the operation of the wind generator was confirmed. We collected frogs from paddy fields (each paddy was about 110 m × 35 m) near wind turbines. At three paddy field sites with independent wind turbines per each area, environmental variables were measured, frogs were collected, and frog sounds were recorded. Each paddy field was with a distance at least 400 to 550 m.

Frogs were considered unsuitable for analysis if they escaped during the sound recording process, they could not be collected after recording, their call note was not clear due to the calling of other frogs or ambient noise during the sound analysis process, there was blood on the swab during the saliva extraction process, or the amount of blood collected from them was insufficient. As a result, 15 frogs were included in analysis for the wind turbine group. Each wind turbine was arranged in a line with a distance of 220–270 m between them in the paddy fields. The wind turbines at this site operated at wind speeds of up to 2 m/s. The wind speed at the time of analysis was 2–4 m/s. All experiments were carried out immediately after the operation of the wind generator was confirmed.

We also assessed the sound pressure of wind turbine noise from directly below the wind turbine (0 m) to 100 m at 20 m intervals on the three sites. The average and standard deviation of sound pressure levels at intervals of 20 m were 57.60 ± 1.11 dB (0 m), 54.23 ± 0.90 dB (20 m), 51.17 ± 1.10 dB (40 m), 47.37 ± 0.76 dB (60 m), 44.53 ± 1.40 dB (80 m), and 41.33 ± 0.45 dB (100 m). In a previous study, the maximum outdoor noise pressure of wind was approximately 55 dBA, while in most studies, the effect of wind noise has been confirmed from an average sound level of over 45 dBA [41]. Thus, we conducted our study within a radius of 80 m of the wind power generator. In addition, since the 80 m radii of each wind power generator did not overlap with each other among the wind power generator intervals (220–270 m), we confirmed the effect of noise on a single wind power generator.

Simultaneously, 24 male frogs were observed in three paddy areas (1.7 to 1.8 km^2^) without wind power generators (control group) from Yeonggwang-gun. As with the wind turbine group, the investigation was conducted at three paddy fields (each paddy was about 110 m × 35 m) per area, and each paddy field was with a distance at least 450 to 500 m. In a previous study, it was reported that low-frequency noise from a wind generator (2.3 MW–2.6 MW) could be detected at 629–1227 m [42]. Therefore, to eliminate the influence of these effects on the study results, the control group was located 2–3 km away from the wind turbine group (Figure 1). Frogs deemed unsuitable for analysis (according to the same criteria outlined for the wind turbine group) were excluded, resulting in the selection of 15 control frogs. One team comprised of three researchers conducted the field investigations, sample collection, and sound recording from 9:00 p.m. to 11:00 p.m. in the wind turbine field, while another identical team conducted the same assessments in the control group fields. One of the team members measured environmental variables, and the other two members recorded the sounds of the frogs and collected the frogs in the nearby site where the environment variables were measured. We recorded the noise under the wind turbine for 5 min three times from each area (a total of three areas) with a wind power generator in Yeonggwang-gun and recorded three values for the air temperature (°C), humidity (%), illuminance (lux), and background noise (dB) at each investigation site (a total of six areas). Each of the sites measuring the environment variables was three sites per area, and they were located at about a 450 to 550 m distance, the same as the investigation site. At each site, measurements were repeated three times at intervals of 50 m, and the value was used after averaging. To obtain the most relevant data, all measurements were collected by placing the equipment as close to the bottom of the paddy field as possible. In addition, the chorus size, which can affect the calling pattern, was measured at each site. The chorus size was confirmed by measuring the number of calling frogs within 5 m × 5 m from the point where the environmental variables were measured. The size of the chorus was also measured three times for each site at 50 m intervals.

### 2.2. Sample Collection in Field

We carefully approached the frogs and recorded the advertisement call for 1 min 30 s using a super directional field recorder and shotgun microphone (ZOOM F1-SP, Tokyo, Japan) at least 1 m away from each frog. The microphone was set to a saving file for 48 k 16 bit, lo-cut and limiter off, recording angle 45°. After the sounds were recorded, the frogs were collected, and their saliva was extracted within 3 min of collection to perform two hormone assays (corticosterone and testosterone). Since the corticosterone can increase rapidly from the stress of capture and handling after 3 min [43], we attempted to analyze the hormone in the saliva from frogs that could be extracted immediately from the fields. The mouth of each individual was opened with a sterile pipette tip or cotton swab, and the saliva sample was extracted using a pre-weighed dry cotton ball for 30 s to 1 min immediately after the collection of a frog. The extracted saliva was stored in 1.5 mL microcentrifuge tubes labeled with the weight of each cotton ball, and all saliva samples were stored in an ice box with ice in the fields. After the saliva was collected, each frog was labeled and kept in a 15 cm bottle with fresh water and breathable pores and immediately transferred to a laboratory. The frogs were anesthetized with ice-cold water, and blood was obtained using heparinized capillary tubes from the cardiac venipuncture to measure the bacterial killing ability (BKA). Considering that the bacterial killing ability can change within 18 to 24 h of collection [44,45], blood was taken within 8 h. Individuals from which blood was extracted were stored in 70% ethanol after pithing to remove body water for body composition analysis using dual energy X-ray absorptiometry (Medikors InAlyzer, Seongnam, Korea), located at the Korea Basic Science Institute (Gwangju, Korea). The procedure for analyzing body composition by dehydration treatment was identical to the previous studies [46], as frogs have permeable skin, so body water content can change rapidly and this can affect the body composition analysis. Experimental procedures on the animals were conducted in accordance with the regulations and approval of the Experimental Animal Ethics Committee of Kongju National University (KNU_2019-01).

The snout-vent length (SVL), an indicator of amphibian body size, and the body weight of dehydrated specimens stored in 70% ethanol were measured to identify that there was no difference in the basic physical conditions of frogs from the sites with and without wind turbines. A digital caliper was used to measure the SVL to 0.01 mm scale, and a digital balance was used to measure the body weight to a 0.01 g scale. Since the call rate may be affected by the energy storage conditions of the body, such as fat mass [47,48], we measured the body composition (fat and lean body content) and bone mineral density (BMD) of *D. japonicus*.

### 2.3. Noise Exposure Experiment in the Field

As with the field investigation, we confirmed the advertisement calling of this frog species in Gongju-si at the end of May to the middle of August in 2020. A total of 15 male frogs (*D. japonicus*) was hand-captured in Gongju-si in July 2020 (breeding season) from inland paddy fields where the animals would have experienced minimal exposure to noise (turbine, traffic, and airplane noise) as there are no wind power generators, airports, and roads in the area. Frogs were deemed unsuitable for analysis if they were not sexually mature, did not call during the recording process, a sufficient amount of blood was not collected, or blood was detected on the cotton swab during the saliva extraction process. As a result, ten frogs were selected for the noise exposure conditions. These frogs were sexually mature males and distinguished by secondary sexual differentiation characteristics, such as the nuptial pad of the front toe and the vocal sac. We then exposed the ten frogs to the single recorded wind turbine noise from Yeonggwang and assessed them pre- and post-exposure.

The saliva of the frogs in the pre-exposure condition was immediately extracted within 3 min of collection according to the same procedures as described above for hormone analysis. After the extraction of the saliva, the blood from the pre-exposed group was immediately extracted in the field by using heparinized capillary tubes from the cardiac venipuncture to analyze their immunity. They were transferred to a plastic container (460 mm × 300 mm × 170 mm) containing clear tap water and a small plastic land. The plastic containers were placed in the shade near the gathering site. After approximately 1 h of acclimatization, the advertisement call of frogs was immediately recorded for 1 min 30 s using the same materials and methods described for the field investigation. Since the background noise collected during the 1 h acclimatization before recording to the end of the experiment was approximately 38–40 dB (similar to the control group in the field investigation), it was judged that no other background noise was present. After the calling was recorded from the pre-exposure condition, the frogs were exposed to a wind turbine noise for 1 h (post-exposure condition) at 55 dB, which was the recorded sound level at the position of frogs with noise exposure in the field investigation. The turbine noise was emitted through a speaker (Bosswiz BOS-N10, Seongnam, Korea), and the noise was emitted as recorded without amplification and/or correction prior to exposure. Advertisement calls of the frogs in the post-exposure condition were then immediately recorded. Because corticosterone has a circadian rhythm [49,50], the frogs were continuously exposed to turbine noise for 24 h in a plastic container, and the saliva was extracted at the same time from the pre- and post-exposure conditions. To compare the immunity between the pre- and post-exposure conditions, blood from the post-exposure condition was extracted immediately after saliva extraction. Additionally, we measured the SVL, body weight, body composition (fat and lean body content), and BMD in the exposure conditions to assess the basic physical conditions of the individuals. The basic physical conditions were compared to identify the difference between field investigation groups. A summary of our experimental design is presented in Figure 2.

### 2.4. Call and Noise Recording and Sound Analysis

A total of 30 advertisement calls from the field investigation groups and 20 advertisement calls (10 pre- and 10 post-) from the noise exposure group were collected to investigate the calling pattern. Raven Pro 1.6 software (Cornell Laboratory of Ornithology, Ithaca, NY, USA) was used to analyze the advertisement calls. We analyzed three characteristics of frog calling: (1) note duration, indicating the length of the note; (2) dominant frequency, representing the frequency with the most energy; and (3) call rate, calculated by the formula [1/note period (note-to-note spacing)], indicating the speed of call. At least 20 note components per individual were analyzed. The complexity or type of advertisement calling has been reported in several frog species [30,51]; however, no studies have reported on the type of advertisement calling of Japanese tree frogs [52,53]. Furthermore, no other types of call notes were observed in the frogs in our study. Therefore, the call notes analyzed were regarded as calls of the same type.

The noise of the wind turbines recorded in the field was also analyzed to verify that their frequency was superimposed on the frequency of the frog calling from the field investigation groups (30 individuals). Wind turbine noise was recorded three times at each area (total of three areas), and the dominant frequency values of at least three frequency areas were obtained. Various types of aerodynamic and low-frequency nacelle mechanical noises were found in the wind turbine noise (Figure 3a). This noise seems to overlap with the calling of frogs (Figure 3b). Thus, we confirmed whether the double blade aerodynamic noise overlapped with the dominant frequency of frogs to identify if turbine noise could affect frogs.

### 2.5. Saliva Extraction and Salivary Hormone Assay

A total of 30 saliva samples from the field investigation groups and 20 from the noise exposure conditions group were collected to analyze the two types of salivary hormones. The cotton ball used for saliva extraction in the field was weighed after being transferred to the lab within 4 h and stored in a microcentrifuge tube in the freezer at −40 °C for two weeks to increase the precipitation of mucin.

Samples were analyzed referring to a previously reported protocol [54]. After freezing, the cotton ball was placed in a microfuge tube with a hole in it and centrifuged at × *g* 8000 rpm for 10 min after washing with 150 µL of ELISA buffer. The solution obtained by centrifugation was immediately processed for ELISA analysis. In some species of previous studies, there was evidence that proteins in the saliva interfered with the analysis of corticosteroids [54]; however, no such proteins have been identified for *D. japonicus*, nor for the testosterone assay. Therefore, we conducted a preliminary analysis to test whether the removal of saliva interference proteins using trichloroacetic acid (TCA) was suitable for the analysis of corticosteroids and testosterone in *D. japonicus*. Considering that a salivary protein was found to interfere with the analyses (Appendix A), TCA-treated samples of corticosterone were used for analysis of salivary corticosterone and testosterone.g

A 96-well corticosterone enzyme-linked immunoassay kit (501320, Cayman Chemical, Ann Arbor, MI, USA) and a 96-well testosterone ELISA kit (582701, Cayman Chemical, Ann Arbor, MI, USA) were used according to the manufacturers’ instructions to analyze the corticosterone and testosterone levels of *D. japonicus*. All experiments were conducted on the same day, and hormones were analyzed using a microplate spectrophotometer (wavelength 412 nm). No samples contaminated with blood were assayed. Samples were developed on three different plates, and the average of three measurements was used. The all-calculation value of the inter-assay coefficient of variation (CV) based on each sample was <14.1%.

### 2.6. Bacterial Killing Assay Using Plasma

The extracted blood was centrifuged at 3000× *g* for 5 min, and the plasma was collected to analyze the immunity of frogs via the BKA assay. The BKA was performed according to the protocol of a previously published study [55]. Briefly, plasma samples were diluted to 1:20 using Amphibian Ringer’s solution (10 µL plasma: 190 µL Ringer’s solution) and mixed with 10 µL of non-pathogenic *Escherichia coli* (Microbio-Logics #24311-ATCC 8739, St Cloud, MN, USA) working solution (approximately 10^4^ microorganisms). Ringer’s solution (210 µL) was used as the negative control, and a mixture of the *E. coli* working solution diluted in 200 µL of Ringer’s solution was used as the positive control. All plasma samples, positive and negative controls, were incubated at 37 °C for 60 min. After incubation, 500 µL of tryptic soy broth was added to all plasma samples and controls. The thoroughly mixed bacterial suspensions were transferred to 96-well plates in 300 µL in duplicate and incubated at 37 °C for 2 h. Finally, the bacterial optical densities were measured every hour using a microplate spectrophotometer (wavelength 600 nm) for a total of four readings. The BKA was evaluated at the beginning of the bacterial exponential growth phase using the formula [(1 − (optical density of sample/optical density of positive control)], which represents the percentage of killed microorganisms in the plasma samples compared to the positive control.

### 2.7. Statistical Analysis

The assumptions of an unpaired *t*-test, a paired *t*-test, a Mann–Whitney U-test, a Wilcoxon rank–sum test, a one-way ANOVA test, and multiple linear regression analyses (residual) were confirmed by Levene’s equal variances test and a Shapiro–wilk normality test. Subsequent analyses all met the assumptions for the test used. Unpaired *t*-tests were used to confirm the differences in environmental conditions (temperature, humidity, illuminance, chorus size, and background noise due to the presence of wind turbines) in the field investigation to compare the frequencies of frog calling and wind turbine noise. A one-way ANOVA test was used to identify differences in SVL, weight, body composition (fat and lean body content), and BMD between field investigation groups (control and wind turbine group) and noise exposure conditions group. Meanwhile, all BKA assay values for the control group, wind turbine group, and pre-exposure conditions were not normally distributed. Therefore, we normalized the data using the normalization formula and log transformation. However, as the applied transformations did not normalize the data even after two types of normalization, the original data that had not been normalization were used in our study. The differences in immunity (BKA) between the control and wind turbine groups were analyzed using a Mann–Whitney U-test, and the differences in immunity between the pre-exposure and post-exposure conditions were analyzed using the Wilcoxon rank–sum test. The calling pattern (note duration, dominant frequency, and call rate) and physiological status (saliva concentration of corticosterone and testosterone) between the control and wind turbine groups were compared using an unpaired *t*-test. The calling pattern and physiology between the pre- and post-exposure conditions were compared using a paired *t*-test. Multiple linear regression was used to identify the behavioral, physiological, and immunological factors (note duration, dominant frequency, call rate, BKA, salivary corticosterone, and salivary testosterone) with the most influence in the field investigation groups. However, since all variables except for corticosterone did not fit the regression model (Appendix B), we only described the results of multiple regression analysis for corticosterone. Statistical analyses were performed using GraphPad Prism version 7.0 for Windows (GraphPad Software, San Diego, CA, USA). All statistical differences were considered significant at *p* < 0.05.

## 3. Results

### 3.1. Environment of Experimental Sites and Basic Physical Information of Frogs

The double blade aerodynamic noise had no statistically significant difference from the frog’s dominant frequency (Figure 4). Based on this lack of difference of frequency, we believe that wind turbine noise can have physiological and behavioral effects on the frogs through overlapping of the blade aerodynamic noise and the frog callings.

The temperature, humidity, and illuminance (not possible to compute a *t*-test because of the same value) were not significantly different between the control and wind turbine groups’ sites. Similarly, there was no significant difference in the chorus size, which indicates the density of males during the breeding season. In contrast, the background noise was found to have a significant difference (*t* _(16)_ = 18.32, *p* < 0.05) between sites with and without wind turbines (Table 1). The SVL, body weight, fat content mass, fat content ratio, lean body content mass, lean body content ratio, and BMD did not differ significantly between the field investigation groups (control and wind turbine group) and noise exposure conditions group (Table 2).

### 3.2. Difference in the Calling Pattern, Steroid Hormones, and Immunity from the Field Investigation Group

There was no statistically significant effect on the note duration and dominant frequency between the control and wind turbine groups, whereas the call rate of frogs in the wind turbine group was statistically significantly higher (*t* _(28)_ = 84.11, *p* < 0.05) than of those in the control group. Meanwhile, the BKA, representing immunity, was statistically significantly lower (U = 60, *p* < 0.05) in the frogs of the wind turbine group than in those of the control group. Corticosterone, which can represent stress or high-energy metabolism, was statistically significantly higher (*t* _(28)_ = 3.62. *p* < 0.05) in the frogs of the wind turbine group than in those of the control group. However, testosterone, a sex-steroid hormone, showed no statistically significant difference between the groups (Figure 5).

### 3.3. Changes in the Calling Pattern, Steroid Hormones, and Immunity Due to Turbine Noise Exposure

There was no statistically significant effect on the note duration and dominant frequency between the pre-exposure group and post-exposure group. However, the call rate statistically significantly increased (*t* _(9)_ = 4.30, *p* < 0.05) after 1 h of turbine noise exposure. Additionally, the BKA, corticosterone, and testosterone had no statistically significant change after 24 h of exposure to wind turbine noise (Figure 6).

### 3.4. Relationship among Calling Pattern, Status of Hormones and Immunity of D. japonicus

Multiple linear regression analysis of corticosterone was shown to have a suitable regression model (Table 3). The note duration, dominant frequency, and testosterone had no significant relationship with salivary corticosterone levels. However, there was a significant correlation between call rate and corticosterone, and an increase in call rate was related to an increase in the saliva concentration of corticosterone. Similarly, BKA also showed a significant correlation with corticosterone. However, in contrast to the call rate, an increase in corticosterone correlated with a decrease in BKA.

## 4. Discussion

In this study, we observed the behavioral, physiological, and immunological responses of Japanese tree frogs caused by wind turbine noise. The background noise of fields with wind turbines was greater than that of fields without wind turbines. Additionally, the aerodynamic turbine noise of the wind power generator overlapped with the frequency of the frog calling. Accordingly, it was confirmed that the wind turbine noise could affect the frogs. The basic physical conditions did not differ in field investigation groups or the noise exposure conditions group. The dominant frequency, note duration, and saliva testosterone levels did not differ in any experimental conditions. The frogs in the turbine noise groups from the field investigation had a higher call rate, higher saliva corticosterone, and lower immunity. Frogs exposed to turbine noise displayed a change in call rate, whereas physiological and immunological changes were not observed. From the results of the multiple regression analysis in frogs from the field investigation, an increase in corticosterone was associated with an increase in call rate and a decrease in BKA.

Previous studies have shown that testosterone concentrations can affect calling behavior or immunity under certain conditions [56,57]; however, there was no difference between paired conditions in our study. Similarly, the energy metabolism and storage status of frogs, such as body size, weight, composition, and/or the chorus size, which can affect male social interactions during reproduction, may also affect calling patterns [58,59,60,61]. However, no significant differences were observed in these variables. Meanwhile, the changes in the call rate were found in both the field investigation and the noise exposure experiment. Some species of male frogs increased the rate of advertisement calling to the same dominant frequency as the noise in response to the noise [28,30,62]. Similarly, our results showed that the calling pattern of male frogs was altered by wind turbine noise. Contrarily, in the exposure experiment, no change was observed in corticosterone or immunity in response to the turbine noise. In certain frog species, acute stress induces an immediate increase in corticosterone responses within 30 min to 1 h [54]. Additionally, chronic stress, such as constant exposure to traffic noise, can cause chronically high levels of corticosterone [26]. However, in our study, exposure to turbine noise for 24 h did not cause this physiological response. We therefore postulate that wind turbine noise may not be the stressor that increases corticosterone by causing the direct response of HPA axis. Meanwhile, frogs collected from paddy fields with wind turbines were found to have high levels of corticosterone and low levels of immunity. These results suggest that turbine noise induces changes in the endocrine and immune systems as processes resulting from behavioral responses, such as change of call rate, rather than direct stress. We therefore predicted a behavioral–physiological–immunological linkage process that governs these responses.

The expected main process is a decrease in immunity due to constant energy investment through a continuous increase in the call rate. Advertisement calling during the breeding season is expensive and requires high energy from male frogs [22]. The mobilization of energy used for this behavior can be achieved through increased corticosterone levels. The release of corticosterone in frogs raises glucose in the blood, which helps with rapid energy mobilization [63]. Many amphibians appear to increase corticosterone levels to meet energetically demanding reproductive behavior during the breeding season [33]. The concentration of corticosterone is important in the calling parameters of male frogs during breeding season [32]; frogs with high call rates have higher corticosterone levels [31]. However, the sustained maintenance of increased corticosteroids can lead to a decrease in immunity due to changes in the energy tradeoff [64,65]. Similarly, an increase in sexual signals often leads to immunosuppression. Representatively, in the immunocompetence handicap hypothesis (ICHH), testosterone, which is a sex-steroid hormone, plays a dual role in mediating immunosuppression, along with the increased sexual signaling [66]. In our study, although there was no difference in testosterone between the groups, it was confirmed that sexual responses, such as an increase in call rate, induce hormonal changes (corticosterone), leading to immunosuppression similar to that described in the ICHH. In addition to testosterone, since there were no differences in body composition that could indicate energy storage states, we believe that the energy used for advertisement calling may have caused a tradeoff in energy used for immunity. Additionally, the increase in call rate and decrease in immunity caused by an increase in corticosterone seems to support our opinion. However, it seems unlikely that this process occurs in a short period of time, as the exposure experiments did not detect these responses. We conducted the field investigations during the middle to the end of the breeding season. Thus, before the frogs were collected, they may have continuously experienced changes in call rates caused by wind turbine noise. Since our study did not attempt to compare these short- and long-term effects, further studies are needed.

In addition to aerodynamic noise, wind power generators are known to emit low-frequency noise or infrasound [8,12,42]. In the case of infrasound from wind power generators, there are few studies that show that they can directly affect health [39], while low-frequency noise can generate physiological changes, such as endocrine disruption and oxidative stress [15,16]. These stressors have the potential to increase the levels of corticosterone and decrease immunity in synergy with the behavioral–physiological–immunological linkage process that we have identified. Although we have not attempted to confirm this in our study, confirmation of these effects, including those induced by infrasound and low-frequency noise, that can be emitted from wind power generators is warranted in future studies. Additionally, we collected frogs during a specific season and set up the paired sites within a relatively small area range for our purposes to minimize other environmental factors. In particular, this was very important because the behavioral, physiological, and immunological contents that we were measuring were contents that changed within a short period of time. Nevertheless, our sample size and replication still may be to be lacking. Additionally, we do not consider the area itself a measurement value but consider each site in the area a measurement value. However, this scope range may be small, so the statistical test may overestimate the degrees of freedom because of the clustered nature of the data. Therefore, additional studies with a wider range and larger study populations are needed.

Most frogs are threatened with extinction worldwide, and one of the main extinction factors is large-scale epidemics within their populations, such as chytrid fungus (*Batrachochytrium dendrobatidis*) and Ranavirus [63,67]. Accordingly, studies are needed to understand the disease epidemiology of frog populations from various perspectives, such as sex, life history, and seasonal patterns. We designed this study to focus on the response of immunity in male frogs. Physiologically, changes in the concentration of sex-steroid hormones during reproductive periods in males increase, suppress, or redistribute immunity ability, thereby increasing the variation range of immunity levels and changing the susceptibility to disease [56,68,69]. Ecologically, male frogs make the advertisement calling, move to find mates and fight to defend a territory, which can lead to a tradeoff of immunocompetence through extreme energy expenditure [70]. Moreover, the greater movement and longer staying in breeding sites of male frogs to find the females can cause frogs to face the risk of more exposure to diseases [22,70,71,72]. Therefore, immunity in males during reproduction can have the potential to determine the overall disease epidemiology of the local frog population. Recognizing the interaction between artificial and natural factors that can influence disease epidemiology can help to monitor the health of frog populations.

## 5. Conclusions

To the best of our knowledge, this is the first study to investigate the effects of wind power generators on frogs. Considering that the observed physiological stress and decrease in immunity did not occur within 24 h, we postulate that the wind turbines are not acute stressors causing direct endocrine response. Meanwhile, the response to the calling pattern was detected immediately after frogs were exposed to the wind turbine noise for 1 h. This difference was also found in the field, and an increase in the call rate was found to correlate with a decrease in immunity via increased corticosterone levels, thereby highlighting a behavioral–physiological–immunological interconnected process. These findings demonstrate how turbine noise has the potential to sufficiently increase the energy investment of male frogs by changing the call rate. However, considering that there was no difference in basic energy or physiological conditions between the two groups, this is expected to be a physiological and immunological response to the energy tradeoff used for calling. Decreased immunity via physiological responses through energy tradeoffs may be synergistic with other stressors or may be exacerbated if sustained over an extended period of time. It is expected that these energy tradeoffs can alter the disease epidemiology of local populations by regulating the balance between reproduction and immunity. Therefore, our findings suggest that structures created for eco-friendly energy generation may have a negative impact on the surrounding ecosystem, thus warranting further confirmatory studies. As such, continuous monitoring of affected habitats can provide useful insights into how species survive in dynamic environments.

## Figures and Tables

**Figure 1 biology-11-00516-f001:**
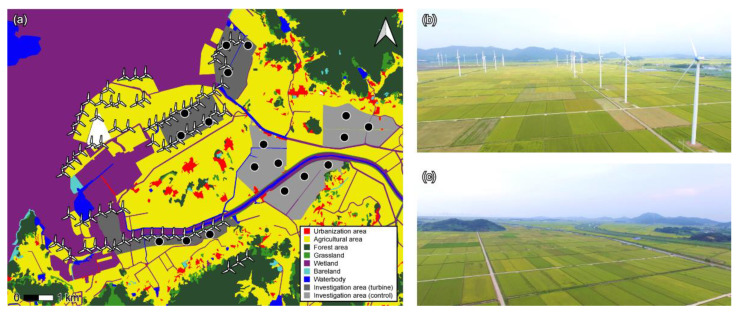
The land cover map with the location of the field investigation in this study and two photos of paddy fields with and without wind power generators, taken by drone. (**a**) Land cover map of field investigation areas. Three investigation sites were contained within each of the three areas with wind power generators (wind turbine group), and the other three investigation sites were contained within each of the three areas without wind power generators (control group). The land cover type of both investigation areas was paddy fields. (**b**) Photo of investigation area with wind turbines (wind turbine group). (**c**) Photo of investigation area without wind turbines (control group).

**Figure 2 biology-11-00516-f002:**
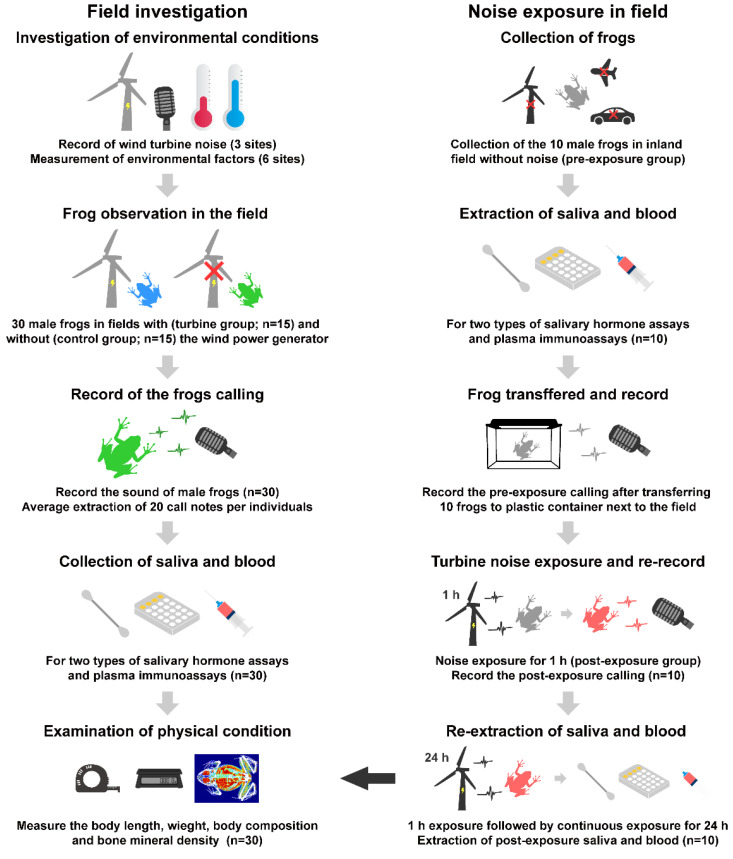
Experimental summary of this study. The figure includes the design of the experiment, the number of frogs collected, and the number of samples used. The field investigation was conducted in Yeonggwang-gun and was conducted in three sites of paddy field areas with wind power generators and three sites of paddy field areas without wind power generators. The noise exposure experiment was conducted in Gongju-si, where the noise of airplanes, traffic, and wind power generators was unprecedented.

**Figure 3 biology-11-00516-f003:**
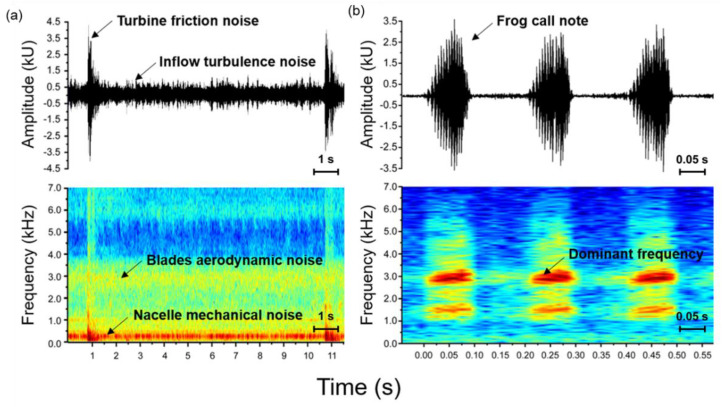
Oscillogram and sound spectrogram of the frog calling and turbine noise. (**a**) Oscillogram (**top**) and sound spectrogram (**bottom**) of the turbine noise in wind power generators. (**b**) Oscillogram (**top**) and sound spectrogram (**bottom**) of the advertisement call in *D. japonicus*. The axis limits of the oscillogram are different for Figure 3a,b.

**Figure 4 biology-11-00516-f004:**
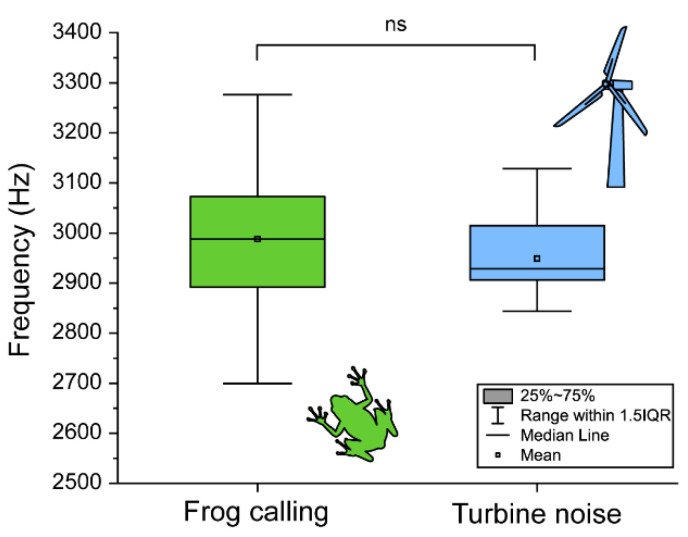
The comparison between the frequency of frog calling (green box) and wind turbine noise (blue box). Box plots show the mean (central square dot), median (central line), 25th and 75th percentiles (bottom and top of boxes), and range within 1.5 interquartile (bottom and top of line).

**Figure 5 biology-11-00516-f005:**
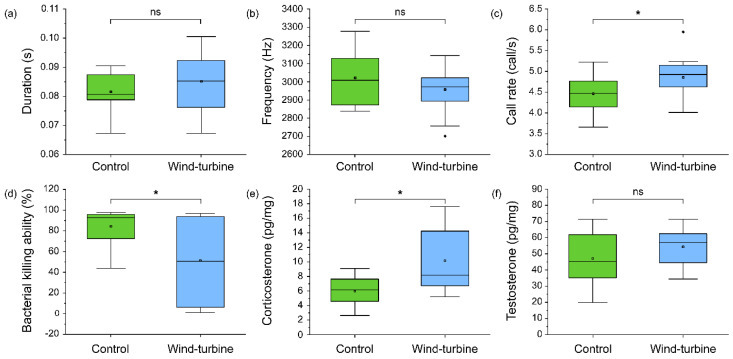
The difference in calling pattern, immunity, and physiological conditions between the control (green box) and wind turbine groups (blue box): (**a**) note duration, (**b**) dominant frequency, (**c**) call rate, (**d**) bacterial killing ability, (**e**) corticosterone, and (**f**) testosterone. Significant differences (*p* < 0.05) were determined using an unpaired *t*-test (and a Mann–Whitney U-test in immunity graph) and are represented by asterisks (*).

**Figure 6 biology-11-00516-f006:**
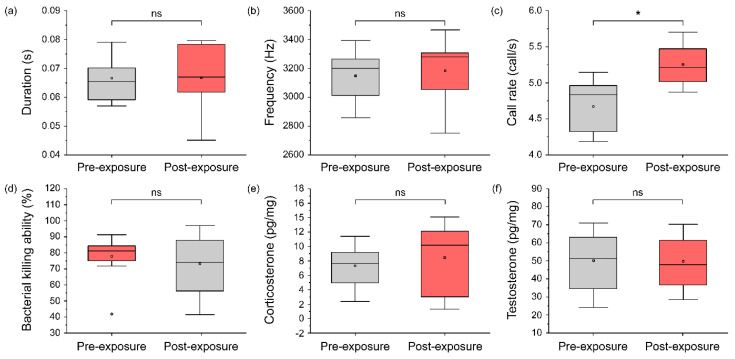
The change in calling pattern, immunity, and physiological conditions due to exposure to wind turbine noise: (**a**) note duration, (**b**) dominant frequency, (**c**) call rate, (**d**) bacterial killing ability, (**e**) corticosterone, and (**f**) testosterone. The gray boxes represent pre-exposure conditions, and the red boxes represent post-exposure conditions. Significant differences (*p* < 0.05) were determined using a paired *t*-test (and Wilcoxon rank–sum test in immunity graph) and are represented by asterisks (*).

**Table 1 biology-11-00516-t001:** The comparison of environmental conditions (temperature, humidity, illuminance, chorus size, and background noise in the presence of wind turbines) between control and wind turbine groups. Significant differences (*p* < 0.05) were determined using an unpaired *t*-test and are represented with asterisks (*).

Contents	Control	Wind Turbine
Temperature (℃)	23.76 ± 0.73	23.75 ± 0.73
Humidity (%)	70.76 ± 3.26	69.27 ± 4.29
Illuminance (lux)	0	0
Chorus size	3.89 ± 1.05	3.56 ± 1.01
Background noise (dB)	40.63 ± 1.55 *	55.77 ± 1.93 *

**Table 2 biology-11-00516-t002:** The comparison of snout-vent length (SVL), body weight, fat content, lean body content, and bone mineral density (BMD) between field investigation groups (control and wind-turbine group) and noise exposure conditions group to identify the differences of basic physical conditions. All contents were not significantly different (*p* > 0.05) according to a one-way ANOVA test.

Contents	Control Group	Wind Turbine Group	Exposure Group
SVL (mm)	30.94 ± 1.22	31.03 ± 1.34	30.59 ± 1.39
Body weight (g)	2.76 ± 0.61	2.83 ± 0.44	2.67 ± 0.77
Fat mass (g)	0.62 ± 0.21	0.78 ± 0.41	0.74 ± 0.26
Fat ratio (%)	22.26 ± 4.25	26.98 ± 11.61	27.18 ± 5.16
Lean mass (g)	2.08 ± 0.42	2.00 ± 0.30	1.91 ± 0.52
Lean ratio (%)	75.99 ± 3.88	71.49 ± 11.18	71.68 ± 4.92
BMD (g/cm^2^)	0.05 ± 0.01	0.05 ± 0.01	0.05 ± 0.01

**Table 3 biology-11-00516-t003:** Analysis of behavioral (note duration, dominant frequency, and call rate); physiological (testosterone, T); and immunological (bacterial killing ability, BKA) factors that significantly interact with corticosterone (CORT) determined using multiple linear regression analysis. Call rate and BKA represents significant correlation (*p* < 0.05) with the CORT.

Predicted Variable	Predictor Variable	Coeff.	Std. Err.	Beta	t	*p*	R^2^	F (*p*)	Df 1, 2
CORT	(Intercept)	−16.41	18.05		−0.91	0.37	0.41	3.36 (0.02)	5, 24
	Duration	77.86	74.22	0.18	1.05	0.31			
	Frequency	<0.01	0.01	0.11	0.68	0.51			
	Call rate	2.72	1.23	0.35	2.21	0.04			
	BKA	−0.06	0.02	−0.52	−3.25	<0.01			
	T	−0.01	0.05	−0.02	−0.13	0.90			

## Data Availability

Not applicable.

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
