# Peer review of "Wind Turbine Noise Behaviorally and Physiologically Changes Male Frogs"

_biology, 2022, doi:10.3390/biology11040516_

Round 1

Reviewer 1 Report

This is a very nice and valuable study with interesting findings. I have no major concerns, apart from the statistical analyses. I appreciate that you have actually made an effort to check the statistical assumptions of the analyses and report this in the manuscript (many authors fail to do that), but I still have some reservations which I think should be addressed in a revision:

L315-318: "Unpaired t-tests" require that each data point is independent from all others. This is not met by your "environmental measurements" because you took 3 repeated (non-independent) samples from each of the 6 sites. The 6 sites are also not fully independent of each other, because 3 sites were around a single wind turbine and 3 other sites were at another location. So these data have a hierarchical correlation structure, which would demand a mixed-modelling approach. However, the sample sizes are so small (i.e. 3 versus 3 sites) that the right model might not be able to provide meaningful estimates. I recommend that you simply present the measured data on boxplots, without statistical comparison.

L318-333: Some level of non-independence may be present in these data, too. The spatial design of the study is not clear (see comment below), but it is possible that the 15 frogs per group may not represent 15 independent data points but cluster into 3 sub-groups (i.e. 3 paddy fields). If this is the case, mixed models would be the ideal method, but I do not think that it would have enough power (i.e. 3 versus 3 sites). Therefore, I recommend that you keep the ANOVAs and t-tests but explicitly state in the manuscript that these tests over-estimate the degrees of freedom because of the clustered nature of the data.

L333-335: This multiple regression analysis should be removed for several reasons. First, it is a fishing experiment with no predictions. It does not add anything that cannot be concluded from Figure 4. Second, it is stated in L336 that the variables did not fulfil the requirements of the statistical method; this means that the results are not reliable. Although model fit was reportedly good for corticosterone, I suspect that you did not check for multi-colinearity as one of the model-fit criteria. From Figure 4, it is assumable that multi-colinearity was present. Third, even if model fit was good, analyzing the data this way is not valid because of pseudoreplication (i.e. the assumption of independence is not met because the frogs sampled from one site are not independent from each other). Mixed models would be needed but they would have extremely low power because there is only N=1 control site and N=1 noise site. (As a side note, the interpretation of the regression results in L397-400 is incorrect, because no interaction was tested but the relationships are phrased as "interactions".)

Minor comments:

L118-161: The design of the field study is not explained clearly enough. First, you write that the study site is "Yeonggwang-gun and Gongju-si". Then, it seems that there were two sites: one with turbines (L131) and one without. Then, it seems there were 3 sites within each site, i.e. 3 rice fields around a single wind power generator (135). The size of either the rice fields, or the total of three fields, was 110 m × 35 m (this is unclear from the text). Then, you write that you measured the chorus size at each site, which were 5 m × 5 m (L161). Please clarify this confusing description that uses "site" for four different things. Also, if possible, please provide a map or drawing to illustrate the spatial arrangement of your study sites in relation to each other and the turbines.

L155-161: Please specify in the text when these measurements were taken relative to each other (e.g. were all measurements taken right after each other?) and to the collection of frogs (e.g. were the environmental data gathered around the same time as the sound recordings?)

L160: Please describe in the text how you measured "chorus size". How do you even define it in a huge continuous area like the ones shown in Figure 1? Where does the chorus end?

L170-180: Please explain in the text why you decided to measure hormones from saliva and not from blood. Is the body size of the frogs too small for allowing enough blood to be taken for both immunity measurement (as in L207) and hormone assays?

L179-181: Is it appropriate to use ethanol-dehydrated specimens for DEXA analysis of body composition?

L184-188: Please clarify in the text if SVL and body mass were measured on living animals or on dehydrated specimens stored in ethanol.

L252-258: I think these sentences should be moved into the Results section.

L263: Why are the axis limits of oscillogram different for Figure 3a and Figure 3b? Why don't you use the same range for both?

L266-267: There are no outliers and no significant differences in this graph, so these lines should be deleted. The information that an independent-samples t-test was used should be moved to L256.

L271: How were the saliva samples stored in the field and how long did it take to get them into the laboratory?

L434-436: Actually, chronic stress can cause higher or lower levels of corticosterone or not change at all: see the 2013 review of Dickens & Romero (Gen. Comp. Endocrinol. 191:177-189).

L437-438: It is unclear what you mean by "direct endocrine disruption". The most parsimonious explanation to your findings is that noise increases corticosterone levels only after a prolonged time (i.e. > 24 h). This does not need to have anything to do with "stress". It may be a consequence of prolonged physical exertion (calling at a high rate, presumably for several months in the field-caught frogs), given the important role of corticosterone in energy mobilization and meeting metabolic needs.

L440: If acclimation was at play, you would not see elevated corticosterone levels in the field-caught frogs at the turbine site.

L462-464: No, the present study did not confirm that increased call rate induces hormonal changes that lead to immunosuppression. This is a likely explanation, but the study was correlational in these respects and thus it did not test any cause–effect relationships (e.g. you did not manipulate corticosterone levels to test if they influence immunity). Please revise your text throughout to avoid such incorrect interpretations (e.g. L33-35, L468).

L484-485: What do you mean by this? The fact that the data are normally distributed does not tell anything about their level of being representative of the population.

Phrasing issues:

I do not think the title is accurate. What the study found in that frogs in wind-turbine sites have higher call rates, higher corticosterone levels, and lower immune defences, and experimental wind turbine noise also increases call rate. Does a high call rate mean "behavioral stress"? I would not think so. And a high corticosterone level does not necessarily mean physiological stress: elevated baseline glucocorticoid levels are a physiologically normal/adequate response to higher energy demands.

L21: It would be more precise to write that you investigated "behavioral, physiological, and immunological responses".

L87-88: The endocrine stress response, i.e. the elevation of glucocorticoids by stimulation of the hypothalamic-pituitary-adrenal axis, is not a "disruption of the endocrine system". This is what a healthy, non-disrupted endocrine system does in response to an acute stressor. Similarly, in L433, it would be better to say that "acute stress induces an immediate increase in saliva corticosterone within 30 min to 1 h"

L96-97: It would be better to say that you studied the "behavioral and physiological responses to wind turbine noise". The word "stress" is too vague in this context.

L107-108: The physiological and immunological responses might have been caused by the noise, or by the behavioral change induced by the noise, or both. Better to say something like: "we studied the physiological and immunological effects of noise".

L112: "or whether a more complex process is occurring" – this is unclear.

 L121: A typo: "which was the middle"

L143: Throughout the text, you are repeatedly using the verb "confirm" in an inappropriate way. For example, here what do you mean by "we confirmed the effect of noise on a single wind power generator"? You must have meant something like "investigated" or "studied" or "measured". Please check and correct this throughout the paper (e.g. L107, 185, 189, 226, 229, 237 etc.)

L441-442: This sentence is non-intelligible.

L478: What is "negative stress"?

L507-508: "we postulate that the wind turbines are not single stressors causing direct endocrine disruption". What do you mean by this? Perhaps you meant "acute stressor" rather than "single"? Plus see the problem with "endocrine disruption" above.

L523-524: This is unclear. Please elaborate what you mean (preferably with citations) or delete this sentence.

Author Response

Reviewer 1

This is a very nice and valuable study with interesting findings. I have no major concerns, apart from the statistical analyses. I appreciate that you have actually made an effort to check the statistical assumptions of the analyses and report this in the manuscript (many authors fail to do that), but I still have some reservations which I think should be addressed in a revision:

→ Thank you for the reviewer's valuable comments. We have tried our best to respond to reviewer comments. 

L315-318: "Unpaired t-tests" require that each data point is independent from all others. This is not met by your "environmental measurements" because you took 3 repeated (non-independent) samples from each of the 6 sites. The 6 sites are also not fully independent of each other, because 3 sites were around a single wind turbine and 3 other sites were at another location. So these data have a hierarchical correlation structure, which would demand a mixed-modelling approach. However, the sample sizes are so small (i.e. 3 versus 3 sites) that the right model might not be able to provide meaningful estimates. I recommend that you simply present the measured data on boxplots, without statistical comparison.

→ We agree with the reviewer's opinion. We have found that there may be confusion in the materials and methods we have described and have made some corrections. We also provided a map to explain this in Figure 1. We also add description to the legend in Figure 2.

P3 L130-144: However, we confirmed that the advertisement calling of this frog species occurred from the end of May in 2020 to the middle of August at Yeonggwang-gun, that is the wind farm site of our field investigation study. Thus, the experiment was conducted in July, which was the middle to end of the breeding season in 2020. We investigated 27 male frogs in three wind farm areas (each investiga-tion areas; about 1.7 to 1.8 km2) with a wind power generator (2.0 MW) from Yeonggwang-gun (wind turbine group). There were 8 to 15 wind turbines in each area.  Each wind turbine was arranged in a line with a distance of 220–270 m between them in the paddy fields. The wind turbines at this site operated at wind speeds of up to 2 m/s. The wind speed at the time of analysis was 2–4 m/s. All experiments were carried out immediately after the operation of the wind genera-tor was confirmed. We collected frogs from paddy fields (each paddy size; about 110 m × 35 m) near wind turbines. In total of 3 paddy fields with independent wind turbines per each area, environmental variable was measured, frogs were collected, and frog sounds were recorded. Each paddy field was with a distance at least 400 to 550m like a wind turbine.

P4 L166-170: Simultaneously, 24 male frogs were observed in three paddy areas (1.7 to 1.8 km2) without wind power generators (control group) from Yeonggwang-gun. As with the wind turbine group, the investigation was conducted at 3 paddy fields (each paddy size; about 110 m × 35 m) per area, and each paddy field was with a distance at least 450 to 550m

P4 L184-187: Each of the sites measuring the environment variables was 3 sites per area, and they were located about 450 to 550m distance same as the investigation site. At each site, measurements were repeated three times at intervals of 50 m, and the value was used after averaging.

P4 L190-192: The chorus size was confirmed by measuring the number of calling frogs within 5 m × 5 m from the point where the environmental variables were measured. The size of the cho-rus was also measured 3 times for each site at 50 m intervals.

→ We considered iterations of 3 sites in one area. In addition, the values of repeated measurements within each site were averaged and used. So, we kept this choice by not considering area as a single measure.

L318-333: Some level of non-independence may be present in these data, too. The spatial design of the study is not clear (see comment below), but it is possible that the 15 frogs per group may not represent 15 independent data points but cluster into 3 sub-groups (i.e. 3 paddy fields). If this is the case, mixed models would be the ideal method, but I do not think that it would have enough power (i.e. 3 versus 3 sites). Therefore, I recommend that you keep the ANOVAs and t-tests but explicitly state in the manuscript that these tests over-estimate the degrees of freedom because of the clustered nature of the data.

→ We decided to keep this because we didn't consider area as a measure, but rather the site within area as a measure. However, in the opinion of the reviewers, this is a small range and seems to have a small number of samples. We also agree on this opinion. Therefore, we described that the clustering characteristics of these places may overestimate the degree of freedom.

P14 L533-541: Additionally, we collected frogs at specific season and set up the paired sites within a relatively small area range, for our purposes to minimize other environmental factors. In particular, this was very important because the behavioral, physiological, and immuno-logical contents that we were measuring were contents that changed within a short period of time. Nevertheless, our sample size and replication still may be to be lacking. Also, we do not consider the area itself as a measurement value, but consider each site in the area as a measurement value. However, this scope range may be small, so the statistical test may over-estimate the degrees of freedom because of the clustered nature of the data.

L333-335: This multiple regression analysis should be removed for several reasons. First, it is a fishing experiment with no predictions. It does not add anything that cannot be concluded from Figure 4. Second, it is stated in L336 that the variables did not fulfil the requirements of the statistical method; this means that the results are not reliable. Although model fit was reportedly good for corticosterone, I suspect that you did not check for multi-colinearity as one of the model-fit criteria. From Figure 4, it is assumable that multi-colinearity was present. Third, even if model fit was good, analyzing the data this way is not valid because of pseudoreplication (i.e. the assumption of independence is not met because the frogs sampled from one site are not independent from each other). Mixed models would be needed but they would have extremely low power because there is only N=1 control site and N=1 noise site. (As a side note, the interpretation of the regression results in L397-400 is incorrect, because no interaction was tested but the relationships are phrased as "interactions".)

Predict the additional results

→ We proceeded with the analysis to draw additional conclusions from Figure 4. This could be meaningful because each individual was tracked identically in sound analysis, hormone analysis, and immunoassay.

Multi-collinearity

→ Multi-collinearity is that shows a strong correlation among independent variables, and can be diagnosed with the coefficient of determination (R2) and the significance probability (p value) of each independent variable. Although the R2 value of the coefficient in determination is high, there are cases where the individual factors are not significant because the p value of each independent variable is large. In this case, one might suspect a high correlation between the independent variables. However, in our corticosterone regression model, a moderate level in coefficient of determination and the F and p values of the model were found to be statistically significant. On the other hand, the p value of each independent variable appeared significantly in some items, call rate and immunity. If the p values in each independent variable were all insignificant, multi-collinearity could be suspected, but since it was not, we judged that this corticosterone model satisfies the regression model assumption. In other regression models, the coefficient of determination was low and the F and p values of the models were not significant, so it was judged that this was not applicable.

Replication and independence:

→ It is likely that there may have been confusion in the statistical analysis due to the problem of sample repetition. In the experimental design, each individual is considered as an independent group of 3 sites in each area. At least 4 frogs were collected from each site. Therefore, we judged that the independence of multiple regression analysis is satisfied. We describe this in detail in the text.

Minor comments:

L118-161: The design of the field study is not explained clearly enough. First, you write that the study site is "Yeonggwang-gun and Gongju-si". Then, it seems that there were two sites: one with turbines (L131) and one without. Then, it seems there were 3 sites within each site, i.e. 3 rice fields around a single wind power generator (135). The size of either the rice fields, or the total of three fields, was 110 m × 35 m (this is unclear from the text). Then, you write that you measured the chorus size at each site, which were 5 m × 5 m (L161). Please clarify this confusing description that uses "site" for four different things. Also, if possible, please provide a map or drawing to illustrate the spatial arrangement of your study sites in relation to each other and the turbines.

→ As mentioned above, this seems to have confused some people. We have summarized this in Materials and Methods and the Figure 1 and Figure 2 legend.

P3 L130-144: However, we confirmed that the advertisement calling of this frog species occurred from the end of May in 2020 to the middle of August at Yeonggwang-gun, that is the wind farm site of our field investigation study. Thus, the experiment was conducted in July, which was the middle to end of the breeding season in 2020. We investigated 27 male frogs in three wind farm areas (each investiga-tion areas; about 1.7 to 1.8 km2) with a wind power generator (2.0 MW) from Yeonggwang-gun (wind turbine group). There were 8 to 15 wind turbines in each area. Each wind turbine was arranged in a line with a distance of 220–270 m between them in the paddy fields. The wind turbines at this site operated at wind speeds of up to 2 m/s. The wind speed at the time of analysis was 2–4 m/s. All experiments were carried out immediately after the operation of the wind genera-tor was confirmed. We collected frogs from paddy fields (each paddy size; about 110 m × 35 m) near wind turbines. At three paddy fields with independent wind turbines per each area, environmental variables were measured, frogs were collected, and frog sounds were recorded. Each paddy field was with a distance at least 400 to 550m like a wind turbine.

P4 L166-170: Simultaneously, 24 male frogs were observed in three paddy areas (1.7 to 1.8 km2) without wind power generators (control group) from Yeonggwang-gun. As with the wind turbine group, the investigation was conducted at 3 paddy fields (each paddy size; about 110 m × 35 m) per area, and each paddy field was with a distance at least 450 to 500m.

P4 L184-187: Each of the sites measuring the environment variables was 3 sites per area, and they were located about 450 to 550m distance same as the investigation site. At each site, measurements were repeated three times at intervals of 50 m, and the value was used after averaging.

P4 L190-192: The chorus size was confirmed by measuring the number of calling frogs within 5 m × 5 m from the point where the environmental variables were measured. The size of the cho-rus was also measured 3 times for each site at 50 m intervals.

L155-161: Please specify in the text when these measurements were taken relative to each other (e.g. were all measurements taken right after each other?) and to the collection of frogs (e.g. were the environmental data gathered around the same time as the sound recordings?)

→ We have included these in the section of material and methods, and have written them in more detail.

P4 L175-180: One team, comprising three researchers, conducted the field investigations, sample collection, and sound recording from 9 pm to 11 pm in the wind turbine field, while another identical team conducted the same assessments in the control group fields. One of the team members measured environmental variables, and the other two members recorded the sounds of frogs and collected the frogs, in the nearby site where the environment variables is measured.

L160: Please describe in the text how you measured "chorus size". How do you even define it in a huge continuous area like the ones shown in Figure 1? Where does the chorus end?

→ We have detailed how to measure the chorus size.

P4 L190-192: The chorus size was confirmed by measuring the number of calling frogs within 5 m × 5 m from the point where the environmental variables were measured. The size of the cho-rus was also measured 3 times for each site at 50 m intervals.

L170-180: Please explain in the text why you decided to measure hormones from saliva and not from blood. Is the body size of the frogs too small for allowing enough blood to be taken for both immunity measurement (as in L207) and hormone assays?

→ We could have drawn enough blood for hormone analysis, but we didn't. The reason for the hormone analysis in saliva was written in detail.

P5 L209-211: Since the corticosterone can increase rapidly by the stress of capture and handling after 3 min [43], we attempted to analyze the hormone in saliva that can be extracted immediately in fields.

L179-181: Is it appropriate to use ethanol-dehydrated specimens for DEXA analysis of body composition?

→ We describe this in the main text.

P5 L223-226: The procedure for analyzing body composition by dehydration treatment was identical to the previous studies [46], as frogs have permeable skin, so body water content can change rapidly and this can affect the body composition analysis.

L184-188: Please clarify in the text if SVL and body mass were measured on living animals or on dehydrated specimens stored in ethanol.

→ We describe this in the main text.

P5 L229-232: The snout-vent length (SVL), an indicator of amphibian body size, and the body weight of dehydrated specimens stored in 70% ethanol were measured to confirm that there was no difference in the basic physical conditions of frogs from the sites with and without wind turbines.

L252-258: I think these sentences should be moved into the Results section.

→ We moved this sentence and figure into the results section.

P10 L388-390: The double blade aerodynamic noise was overlapped (t (37) = 0.816, p = 0.420) with the frog's dominant frequency (Figure 4). Based on this overlapped frequency, we believe that wind turbine noise can have physiological and behavioral effects on the frogs.

L263: Why are the axis limits of oscillogram different for Figure 3a and Figure 3b? Why don't you use the same range for both?

→ The reason the axis limits are different in the oscillogram is because the durations of the two sound are very different. We made this choice because we wanted to show the sounds that make up the wind turbine noise and the frog calling

L266-267: There are no outliers and no significant differences in this graph, so these lines should be deleted. The information that an independent-samples t-test was used should be moved to L256.

→ We removed the description of outliers and the statement that we used the t-test from the figure legend.

L271: How were the saliva samples stored in the field and how long did it take to get them into the laboratory?

→ We have described about this in detail in the main text.

P5 L214-215: The extracted saliva was stored in 1.5ml microcentrifuge tubes written the weighed of each cotton ball, and all saliva samples were stored in an ice box with ice on fields.

P8 L314-317: The cotton ball used for saliva extraction in the field was weighed after being transferred to the lab within 4 h, and stored in a microcentrifuge tube in the freezer at -40 °C for two weeks to increase the precipitation of mucin.

L434-436: Actually, chronic stress can cause higher or lower levels of corticosterone or not change at all: see the 2013 review of Dickens & Romero (Gen. Comp. Endocrinol. 191:177-189).

→ Although not all chronic factors are the result of high corticosterone concentrations, we have discussed similar studies to our study. Although not wind turbine noise, there are instances in traffic noise that indicate high corticosterone concentrations when exposed to continuous noise.

L437-438: It is unclear what you mean by "direct endocrine disruption". The most parsimonious explanation to your findings is that noise increases corticosterone levels only after a prolonged time (i.e. > 24 h). This does not need to have anything to do with "stress". It may be a consequence of prolonged physical exertion (calling at a high rate, presumably for several months in the field-caught frogs), given the important role of corticosterone in energy mobilization and meeting metabolic needs.

→ We strongly agree with the reviewer's comments. We anticipate an increase in corticosterone due to an increase in the long-term increase of call rate. Perhaps this increase seems to be for energy mobilization.  However, in a few previous studies, there have been studies showing that noise itself increases corticosterone by acting as a stressor. We have described this in the introduction and review.

P2 L74-82: Grayleg goose (Anser anser) raised near wind turbines were physiologically affected by the noise itself, so they had higher cortisol level in blood and lower weight gain than geese raised away from turbines. [19]. Turbine noise also affects the vocal communication of Greater prairie chicken (Tympanuchus cupido pinnatus) by changing the soundscape of its surroundings [20]. Additionally, turbine noise actually induced increases in cortisol through the hypothalamic-pituitary-adrenal (HPA) axis response of European badger (Meles meles), which the researchers suggest could alter the risk of infection or disease in the population by changing the immune system [21].

P3 L97-99: Additionally, noise can act directly affects the endocrine system of frogs [26,34]. Change of the endocrine response due to stress leads to increased corticosterone by stimulation of the HPA axis [33,35].

P13 L486-487: Additionally, chronic stress, such as constant exposure to traffic noise, can cause chronically high levels of corticosterone [26].

→ The word "direct endocrine disruption" that we used of means that noise itself does not act as a stressor as in previous studies. We changed the word choice as this could be confusing.

P13 L488-490: We, therefore, postulate that wind turbine noise may not be the stressor that increase in corticosterone by causing the direct response of HPA axis.

→ Agreeing with the reviewer opinion, our key finding is that, although noise can only affect long-term effects by changing call rates. Additionally, we think, another key finding is that, wind-generated noise does not trigger endocrine responses (directly increase of glucocorticoid) that directly transverse the hypothalamic-pituitary-adrenal axis. If so, our noise exposure experiments would have resulted in an immediate increase in corticosterone.

→ We thought that the weak and persistent wind turbine noise would not show a direct corticosterone increase as the previous study did. The sound pressure level (SPL) used in most of the reported studies for traffic noise was around 70-80 dB [26] [Grace and Noss, 2018; Leon et al., 2019; Yi and Sheridan, 2019], and the SPL used in the study for airplane noise was about 80 dB [28]. On the other hands, in our study, the SPL of the wind turbine noise was about 55 dB, which was weaker than that of the two types of noise. Also, wind turbine noise has a continuous noise as the turbine works, rather than having a momentarily strong and irregular pulses [14].  But we didn't include this in our discussion because we didn't compare them directly.

Leon, E., Peltzer, P. M., Lorenzon, R., Lajmanovich, R. C., & Beltzer, A. H. (2019). Effect of traffic noise on Scinax nasicus advertisement call (Amphibia, Anura). Iheringia. Série Zoologia, 109.

Grace, M. K., & Noss, R. F. (2018). Evidence for selective avoidance of traffic noise by anuran amphibians. Animal Conservation, 21(4), 343-351.

Yi, Y. Z., & Sheridan, J. A. (2019). Effects of traffic noise on vocalisations of the rhacophorid tree frog Kurixalus chaseni (Anura: Rhacophoridae) in Borneo. RAFFLES Bulletin of Zoology, 67.

→ Conservation biologists studying frogs are now trying to find endangered factors of frog, which is why it is of interest to identify whether many of the environmental factors that may affect these animals are direct stress (resulting in increased corticosterone by hypothalamic-pituitary-adrenal axis). Therefore, we consider this discussion to be one of the important contents.

L440: If acclimation was at play, you would not see elevated corticosterone levels in the field-caught frogs at the turbine site.

→ We deleted this sentence in discussion because we determined that it did not explain in our results.

L462-464: No, the present study did not confirm that increased call rate induces hormonal changes that lead to immunosuppression. This is a likely explanation, but the study was correlational in these respects and thus it did not test any cause–effect relationships (e.g. you did not manipulate corticosterone levels to test if they influence immunity). Please revise your text throughout to avoid such incorrect interpretations (e.g. L33-35, L468).

→ We agree with our reviewers. There must have been some confusion in our words. Accordingly, we have revised this section.

P1 L32-35: Thus, turbine noise seems to induce decreased immunity in Japanese tree frogs as an increase in energy investment that triggers a behavioral response rather than acting as a sole physiological response that leads to a directly increase in corticosterone.

P13 L510-513: In our study, although there was no difference in testosterone between the groups, it was confirmed that sexual responses, such as an increase in call rate, induce hormonal changes (corticosterone), leading to immunosuppression, similar to that described in the ICHH.

L484-485: What do you mean by this? The fact that the data are normally distributed does not tell anything about their level of being representative of the population.

→ We got rid of this erroneous part. We also revised this sentence to make the limitations of our study clearer.

P14 L533-541: Additionally, we collected frogs at specific season and set up the paired sites within a relatively small area range, for our purposes to minimize other environmental factors. In particular, this was very important because the behavioral, physiological, and immuno-logical contents that we were measuring were contents that changed within a short period of time. Nevertheless, our sample size and replication still may be to be lacking. Also, we do not consider the area itself as a measurement value, but consider each site in the area as a measurement value. However, this scope range may be small, so the statistical test may over-estimate the degrees of freedom because of the clustered nature of the data.

Phrasing issues:

I do not think the title is accurate. What the study found in that frogs in wind-turbine sites have higher call rates, higher corticosterone levels, and lower immune defences, and experimental wind turbine noise also increases call rate. Does a high call rate mean "behavioral stress"? I would not think so. And a high corticosterone level does not necessarily mean physiological stress: elevated baseline glucocorticoid levels are a physiologically normal/adequate response to higher energy demands.

→ We changed the title according to the reviewer's comments.

L21: It would be more precise to write that you investigated "behavioral, physiological, and immunological responses".

→ We changed this.

L87-88: The endocrine stress response, i.e. the elevation of glucocorticoids by stimulation of the hypothalamic-pituitary-adrenal axis, is not a "disruption of the endocrine system". This is what a healthy, non-disrupted endocrine system does in response to an acute stressor. Similarly, in L433, it would be better to say that "acute stress induces an immediate increase in saliva corticosterone within 30 min to 1 h"

→ We changed these sentence.

P3 L98-99: Change of the endocrine response due to stress leads to increased corticosterone by stimu-lation of the HPA axis

P13 L484-485: In certain frog species, acute stress induces an immediate increase in corticosterone responses within 30 min to 1 h.

L96-97: It would be better to say that you studied the "behavioral and physiological responses to wind turbine noise". The word "stress" is too vague in this context.

→ We changed this sentence.

P3 L106-108: In the current study, we investigated the behavioral and physiological responses of male Japanese tree frogs (Dryophytes japonicus) caused by wind turbine noise in paddy fields.

L107-108: The physiological and immunological responses might have been caused by the noise, or by the behavioral change induced by the noise, or both. Better to say something like: "we studied the physiological and immunological effects of noise".

 → We changed this sentence.

P3 L119-120: Finally, we studied the response of steroid hormones and immune system to turbine noise.

L112: "or whether a more complex process is occurring" – this is unclear.

 → We deleted this sentence.

 P3 L122-124: Field investigation and noise exposure analysis of paddy fields were conducted in parallel to identify whether the aerodynamic noise of wind turbines alone cause behavioral and physiological responses.

 L121: A typo: "which was the middle"

 → We revised this typo 

P3 L132-133: Thus, the experiment was conducted in July, which was the middle to end of the breeding season in 2020.

L143: Throughout the text, you are repeatedly using the verb "confirm" in an inappropriate way. For example, here what do you mean by "we confirmed the effect of noise on a single wind power generator"? You must have meant something like "investigated" or "studied" or "measured". Please check and correct this throughout the paper (e.g. L107, 185, 189, 226, 229, 237 etc.)

→ We have corrected this word to be more clearer.

P3 L119-120: Finally, we studied the response of steroid hormones and immune system to turbine noise.

P5 L229-232: The snout-vent length (SVL), an indicator of amphibian body size, and the body weight of dehydrated specimens stored in 70% ethanol were measured to identify that there was no difference in the basic physical conditions of frogs from the sites with and without wind turbines.

P6 L233-236: Since the call rate may be affected by the energy storage conditions of the body, such as fat mass [47,48], we measured the body composition (fat and lean body contents), and bone mineral density (BMD) of D. japonicus.

P6 L273-275: Additionally, we measured the SVL, body weight, and body composition (fat and lean body contents), and BMD in the exposure conditions to assess the basic physical conditions of the individuals.

P6 L275-277: The basic physical conditions were compared to identify the difference with field investigation groups.

P7 L286-288: Thirty advertisement calls of the field investigation groups and 20 advertisement calls (10 pre- and 10 post-) of the noise exposure conditions were collected to investigate the calling pattern.

L441-442: This sentence is non-intelligible.

→ We removed this contents because we felt it was too broad an interpretation for our results.

L478: What is "negative stress"?

→ We modified this word to be a physiological change.

P14 L525-528: In the case of infrasound from wind power generators, there are few studies that show that they can directly affect health [39], while low-frequency noise can generate physiological changes, such as endocrine disruption and oxidative stress

L507-508: "we postulate that the wind turbines are not single stressors causing direct endocrine disruption". What do you mean by this? Perhaps you meant "acute stressor" rather than "single"? Plus see the problem with "endocrine disruption" above.

→ As mentioned above, we wanted to point out that wind turbine noise may be not a direct HPA axis response factor that increases corticosterone. However, it was judged that there was confusion in the word, and the word was corrected according to the opinions of the reviewers.

P14 L562-564: Considering that the observed physiological stress and decrease in immunity did not occur within 24 h, we postulate that the wind turbines are not acute stressors causing direct endocrine response.

L523-524: This is unclear. Please elaborate what you mean (preferably with citations) or delete this sentence.

→ We have removed this sentence.

Reviewer 2 Report

Wind energy although regarded as “green” energy, have negative impacts on different taxa such as birds and bats due to their fatal collisions with the wind turbines. In recent years, the effect of noise generated by wind turbines on wildlife also receive much attention due to its negative impacts on the physiology and behavior of different tax. Thus, the idea of the current research may have numerous implications for conservation of frogs and other wildlife. However, my main concern is not related to the goals of the research but rather related to the scientific attitude of the researchers.

For example, the use the DXA method to measure lean and fat mass of the frogs. This method was developed for measurements of bone mineral content and density in human and adjusted for use in other animals such as birds, lizards and rodents. However, all these studies were done after carful calibration, and validation of the results. Using the DXA without any validation that should be published is unsatisfactory. Along the MS there are several more examples where the authors present data without any explanation about the way that they obtained them or without any scientific judgment.            

Other major comments:

The writing is hard to follow and the MS needs English editing.

The introduction is not focus. The authors present information on offshore turbines or effect of turbine noise on humans, all are less relevant for the current topic. Studies on the effects turbine noise and wildlife are hardly discussed.

Hypothesis and predictions are missing.

Author Response

Wind energy although regarded as “green” energy, have negative impacts on different taxa such as birds and bats due to their fatal collisions with the wind turbines. In recent years, the effect of noise generated by wind turbines on wildlife also receive much attention due to its negative impacts on the physiology and behavior of different tax. Thus, the idea of the current research may have numerous implications for conservation of frogs and other wildlife. However, my main concern is not related to the goals of the research but rather related to the scientific attitude of the researchers.

 → Thank you for the reviewer's positive comments. We did our best to reflect the reviewer comments.

For example, the use the DXA method to measure lean and fat mass of the frogs. This method was developed for measurements of bone mineral content and density in human and adjusted for use in other animals such as birds, lizards and rodents. However, all these studies were done after carful calibration, and validation of the results. Using the DXA without any validation that should be published is unsatisfactory. Along the MS there are several more examples where the authors present data without any explanation about the way that they obtained them or without any scientific judgment.            

→ The DXA we used is an X-ray absorptiometry for small animals, not for humans. Additionally, we previously used the same dual X-ray absorptiometry to analyze the body composition of frogs and to study environmental differences such as trophic levels, nutritional status, predation pressure, and captive environment, including the target species we studied, and to establish a reference interval for diagnosing frog individuals did research. Before carrying out these studies, we reviewed the existing literature on X-ray analysis for small animals, compared the measured weight and actual weight, confirmed the error range according to repeated measurement of X-ray, and identified the minimum weight that is stable for dual X-ray absorptiometry. Accordingly, we tested and verified whether this dual X-ray absorptiometry is suitable for analyzing the body composition of frogs.

Park, Jun Kyu, and Yuno Do. "Assessment of body condition in amphibians using radiography: Relationship between bone mineral density and food resource availability." Korean Journal of Ecology and Environment 52.4 (2019): 358-365.

Park, Jun-Kyu, and Yuno Do. "Evaluating the physical condition of Hyla japonica using radiographic techniques." Science of The Total Environment 726 (2020): 138596.

Park, Jun-Kyu, Jeong-Bae Kim, and Yuno Do. "Reference intervals in combined veterinary clinical examinations of male black-spotted pond frogs (Pelophylax nigromaculatus)." Animals 11.5 (2021): 1407.

Park, Jun-Kyu, Jeong Bae Kim, and Yuno Do. "Examination of Physiological and Morphological Differences between Farm-Bred and Wild Black-Spotted Pond Frogs (Pelophylax nigromaculatus)." Life 11.10 (2021): 1089.

Other major comments:

The writing is hard to follow and the MS needs English editing.

→ We entrusted the English proofing of the manuscript to native speakers.

The introduction is not focus. The authors present information on offshore turbines or effect of turbine noise on humans, all are less relevant for the current topic. Studies on the effects turbine noise and wildlife are hardly discussed.

→ We have discussed in more detail the impact on wildlife in the introduction.

P2 L55-58: Accordingly, the impact of wind power generation on marine ecosystems is an emerging concern, prompting studies focusing on the impact of habitat degradation, noise exposure response, and reduced reproductive capacity on marine organisms, such as marine mammals, fishes, birds, and algae [10-13].

P2 L74-82: Grayleg goose (Anser anser) raised near wind turbines were physiologically affected by the noise itself, so they had higher cortisol level in blood and lower weight gain than geese raised away from turbines. [19]. Turbine noise also affects the vocal communication of Greater prairie chicken (Tympanuchus cupido pinnatus) by changing the soundscape of its surroundings [20]. Additionally, turbine noise actually induced increases in cortisol through the hypothalamic-pituitary-adrenal (HPA) axis response of European badger (Meles meles), which the researchers suggest could alter the risk of infection or disease in the population by changing the immune system [21].

Hypothesis and predictions are missing.

→ We have written in detail the hypotheses and predictions in the introduction.

P3 L113-115: We assumed that the wind turbine noise could change the pattern of frog calling. It was also expected that this would alter the physiological state for energy mobilization and lead to a decrease in immunity.

Reviewer 3 Report

This is a well written paper which gives useful information on the impacts of wind turbines on frogs. This is the first paper I have read on this topic, the findings will be of international interest.

The only area of the paper where I thought the writing could be improved is in the statistical methods section. The sentence "Subsequent analyzes were selected based on confirmed assumption." is poor may I suggest Subsequent analyses all met the assumptions for the test used.  In this section it seems the data could not be normalised using common transformations  so the authors used non-parametric statistical tests. This all seems fine. If they wanted to try a different approach they might consider using generalized linear modelling which would avoid the need to initially transform  the data. However, I think their approach is fine, if a little old fashioned.  Another sentence that can be improved begins "However, since the data does not have a normality distribution even..." I would suggest.. as the applied transformations did not normalise the data...

Author Response

Reviewer 3

This is a well written paper which gives useful information on the impacts of wind turbines on frogs. This is the first paper I have read on this topic, the findings will be of international interest.

The only area of the paper where I thought the writing could be improved is in the statistical methods section. The sentence "Subsequent analyzes were selected based on confirmed assumption." is poor may I suggest Subsequent analyses all met the assumptions for the test used.  In this section it seems the data could not be normalised using common transformations so the authors used non-parametric statistical tests. This all seems fine. If they wanted to try a different approach they might consider using generalized linear modelling which would avoid the need to initially transform  the data. However, I think their approach is fine, if a little old fashioned.  Another sentence that can be improved begins "However, since the data does not have a normality distribution even..." I would suggest. as the applied transformations did not normalise the data...

→ Thank you for the reviewer's valuable comments. We have tried our best to respond to reviewer comments. We corrected the sentence according to comment of reviewer.

P9 L359-360: Subsequent analyses all met the assumptions for the test used.

P9 L369-371: However, as the applied transformations did not normalize the data even after two types of normalization, the original data that has not been normalization was used in our study.

Reviewer 4 Report

In the manuscript entitled “Wind turbine noise behaviorally and physiologically stresses male frogs”, the authors analyzed the behavioral-physiological-immunological interconnected process of Japanese tree frogs (Dryophytes japonicus) during the breeding season when exposed to wind turbine noise. This is the first study to investigate the effects of wind power generators on frogs and could provide insights into the potential negative impact that the eco-friendly energy generation may have on the surrounding ecosystem. In the followings, there are some comments that the authors need to notice.

  1. As the authors mentioned (at line 485 of the manuscript), the sample sizes are small in this study. The field investigation was conducted in only three sites with a wind power generator and three sites without it. Comparing more sampling sites are needed to increase the strength of scientific evidence supporting the main point of this study that wind turbine noise stresses male frogs. In addition, the results in this study showed that the wind turbine group in the field had a higher call rate and corticosterone level, and lower immunity than the group in the field without turbines present, which are not consistent to those in the noise exposure experiment that a higher call rate was only observed post-exposure compared to pre-exposure and no differences on the corticosterone level or the immunity reaction were found. The inconsistent results indicate that more direct and powerful evidences are necessary to verify whether turbine noise stresses male frogs. Many variables (such as using of herbicides and fertilizers, pH value, soil type, abundance and diversity of prey and predators, parasites, pathogen and disease, wind velocity, ground vibration, human activity, etc.) may also cause the physiological and immunological differences between the two groups and need to be verified before the conclusion is made. The authors also need to sample at more time points (besides 24 h) in the noise exposure experiment to correctly verify the direct effects of noise exposure on frogs.
  2. At line 122, how far were the rice paddy fields located away from each other?
  3. At lines 129, 148, and 199, how to “randomly” select the frogs?
  4. At line 156, how far were the sampling places for the three values of environmental variables located away and how to randomly select the sampling places? In addition, the three values for each variable should be nested under the site factor in statistics, and it may cause pseudoreplication problems if the sample size for each group is calculated as 3 values multiplied by three sites (n = 9). The values per site should be averaged and the sample size for each group will be only n = 3. Therefore, the df values for t-tests at lines 343-346 should be corrected as 4.
  5. At line 160, how to determine the chorus size?
  6. At line 256, why is the df value of this t-test presented as 37?
  7. At lines 262-263, “D. japonicus” should be italicized.
  8. At line 265, “central band” should be revised as “central line”?
  9. At lines 266-267, the description “, outlier (diamonds dots). Significant differences (p < 0.05) were determined using an unpaired t-test and are represented by asterisks (*)” is redundant because neither “outlier” nor “p < 0.05” are presented in the figure 3(c), where the legend should also be revised.
  10. At lines 283, 337, “Appendix 1” and “Appendix 2” should be revised as “Appendix A” and “Appendix B”, respectively.
  11. At line 405, “corticosterone” should be revised a “corticosterone (CORT)”.
  12. At line 427, “changes in the call rate in the field investigation and noise exposure experiment” is redundant and should be removed.
  13. At line 441, “occur” should be revised.
  14. At line 446, what are the behavioral responses?
  15. At line 455, as we know that the concentration of corticosterone is important in the calling parameters of male frogs, why did the corticosterone level not increase with call rate in the noise exposure experiment in your study?

Author Response

Reviewer 4

In the manuscript entitled “Wind turbine noise behaviorally and physiologically stresses male frogs”, the authors analyzed the behavioral-physiological-immunological interconnected process of Japanese tree frogs (Dryophytes japonicus) during the breeding season when exposed to wind turbine noise. This is the first study to investigate the effects of wind power generators on frogs and could provide insights into the potential negative impact that the eco-friendly energy generation may have on the surrounding ecosystem. In the followings, there are some comments that the authors need to notice.

→ Thank you for the reviewer's valuable comments. We did our best to reflect the opinions of the reviewers and revise our manuscript.

As the authors mentioned (at line 485 of the manuscript), the sample sizes are small in this study. The field investigation was conducted in only three sites with a wind power generator and three sites without it. Comparing more sampling sites are needed to increase the strength of scientific evidence supporting the main point of this study that wind turbine noise stresses male frogs. In addition, the results in this study showed that the wind turbine group in the field had a higher call rate and corticosterone level, and lower immunity than the group in the field without turbines present, which are not consistent to those in the noise exposure experiment that a higher call rate was only observed post-exposure compared to pre-exposure and no differences on the corticosterone level or the immunity reaction were found. The inconsistent results indicate that more direct and powerful evidences are necessary to verify whether turbine noise stresses male frogs. Many variables (such as using of herbicides and fertilizers, pH value, soil type, abundance and diversity of prey and predators, parasites, pathogen and disease, wind velocity, ground vibration, human activity, etc.) may also cause the physiological and immunological differences between the two groups and need to be verified before the conclusion is made. The authors also need to sample at more time points (besides 24 h) in the noise exposure experiment to correctly verify the direct effects of noise exposure on frogs.

→ We have added the contents of sample size in discussion section. Also, instead of measuring many variables, we narrowed the temporal and spatial scale as much as possible. Of course, it could have been better to extend the temporal and spatial range and measure multiple variables. But we couldn't do that, so we added this to the discussion.

P14 L533-541: Additionally, we collected frogs at specific season and set up the paired sites within a relatively small area range, for our purposes to minimize other environmental factors. In particular, this was very important because the behavioral, physiological, and immuno-logical contents that we were measuring were contents that changed within a short period of time. Nevertheless, our sample size and replication still may be to be lacking. Also, we do not consider the area itself as a measurement value, but consider each site in the area as a measurement value. However, this scope range may be small, so the statistical test may over-estimate the degrees of freedom because of the clustered nature of the data.

→ We also added to the discussion the need to check long-term and short-term reactions.

P14 L520-523: Thus, before the frogs were collected, they may have continuously experienced changes in call rates caused by wind turbine noise. Since our study did not attempt to compare these short- and long-term effects, further studies are needed.

→ Nevertheless, we think our study is worthwhile as it is the first study of frogs on wind power noise and (as far as we know) one of the few studies that analyzed the impact of offshore wind farms from the perspective of freshwater ecosystems.

At line 122, how far were the rice paddy fields located away from each other?

→ We added this content to Figure 1.

At lines 129, 148, and 199, how to “randomly” select the frogs?

→ We did not "randomly" select the frogs we sampled. As noted in the text, frogs not suitable for analysis were excluded. We tried to use "all" of the frogs we sampled by selecting the frogs we used for the analysis.

P3-4 L145-149: Frogs were considered unsuitable for analysis if the frogs escaped during the sound recording process, the frogs could not be collected after recording, the call note of the frogs was not clear due to the calling of other frogs or ambient noise during the sound analysis process, there was blood on the swab during the saliva extraction process, or the amount of blood collected from the frogs was insufficient.

P4 L173-175: Frogs deemed unsuitable for analysis (according to the same criteria outlined for the wind turbine group) were excluded, resulting in the selection of 15 control frogs.

P6 L243-246: Frogs were deemed unsuitable for analysis if they were not sexually mature, did not call during the recording process, a sufficient amount of blood was not collected, or blood was detected on the cotton swab during the saliva extraction process.

At line 156, how far were the sampling places for the three values of environmental variables located away and how to randomly select the sampling places? In addition, the three values for each variable should be nested under the site factor in statistics, and it may cause pseudoreplication problems if the sample size for each group is calculated as 3 values multiplied by three sites (n = 9). The values per site should be averaged and the sample size for each group will be only n = 3. Therefore, the df values for t-tests at lines 343-346 should be corrected as 4.

→ We agree with the reviewer's opinion. We have found that there may be confusion in the materials and methods we have described and have made some corrections. We also provided a map to explain this in Figure 1. We also add description to the legend in Figure 2.

P3 L130-144: However, we confirmed that the advertisement calling of this frog species occurred from the end of May in 2020 to the middle of August at Yeonggwang-gun, that is the wind farm site of our field investigation. Thus, the experiment was conducted in July, which was the middle to end of the breeding season in 2020. We investigated 27 male frogs in three wind farm areas (each investiga-tion areas; about 1.7 to 1.8 km2) with a wind power generator (2.0 MW) from Yeonggwang-gun (wind turbine group). There were 8 to 15 wind turbines in each area. Each wind turbine was arranged in a line with a distance of 220–270 m between them in the paddy fields. The wind turbines at this site operated at wind speeds of up to 2 m/s. The wind speed at the time of analysis was 2–4 m/s. All experiments were carried out immediately after the operation of the wind genera-tor was confirmed. We collected frogs from paddy fields (each paddy size; about 110 m × 35 m) near wind turbines. At three paddy fields with independent wind turbines per each area, environmental variables were measured, frogs were collected, and frog sounds were recorded. Each paddy field was with a distance at least 400 to 550m like a wind turbine.

P4 L166-170: Simultaneously, 24 male frogs were observed in three paddy areas (1.7 to 1.8 km2) without wind power generators (control group) from Yeonggwang-gun. As with the wind turbine group, the investigation was conducted at 3 paddy fields (each paddy size; about 110 m × 35 m) per area, and each paddy field was with a distance at least 450 to 500m.

P4 L184-187: Each of the sites measuring the environment variables was 3 sites per area, and they were located about 450 to 550m distance same as the investigation site. At each site, measurements were repeated three times at intervals of 50 m, and the value was used after averaging.

P4 L190-192: The chorus size was confirmed by measuring the number of calling frogs within 5 m × 5 m from the point where the environmental variables were measured. The size of the cho-rus was also measured 3 times for each site at 50 m intervals.

→ We considered iterations of 3 sites in one area. In addition, the values of repeated measurements within each site were averaged and used. So, we kept this choice by not considering area as a single measure.

At line 160, how to determine the chorus size?

→ We have detailed how to measure the chorus size.

P4 L190-192: The chorus size was confirmed by measuring the number of calling frogs within 5 m × 5 m from the point where the environmental variables were measured. The size of the cho-rus was also measured 3 times for each site at 50 m intervals.

At line 256, why is the df value of this t-test presented as 37?

→ We have described an explanation for this.

P8 L298-302: The noise of the wind turbines recorded in the field was also analyzed to verify that their frequency was superimposed on the frequency of the frog calling from the field investigation groups (30 individuals). Wind turbine noise was recorded three times at each area (total of three area), and the dominant frequency values of at least three frequency areas were obtained.

At lines 262-263, “D. japonicus” should be italicized.

→ We revised this typo.

At line 265, “central band” should be revised as “central line”?

→ We revised this.

At lines 266-267, the description “, outlier (diamonds dots). Significant differences (p < 0.05) were determined using an unpaired t-test and are represented by asterisks (*)” is redundant because neither “outlier” nor “p < 0.05” are presented in the figure 3(c), where the legend should also be revised.

→ We removed the description of outliers from the figure legend and deleted the statement that we used the t-test. Additionally, we moved this to the results section.

P10 L388-390: The double blade aerodynamic noise was overlapped (t (37) = 0.816, p = 0.420) with the frog's dominant frequency (Figure 4). Based on this overlapped frequency, we believe that wind turbine noise can have physiological and behavioral effects on the frogs.

At lines 283, 337, “Appendix 1” and “Appendix 2” should be revised as “Appendix A” and “Appendix B”, respectively.

→ We revised this typo.

At line 405, “corticosterone” should be revised a “corticosterone (CORT)”.

→ We revised this.

At line 427, “changes in the call rate in the field investigation and noise exposure experiment” is redundant and should be removed.

→ We removed the redundant sentences.

At line 441, “occur” should be revised.

→ We removed this contents because we felt it was too broad an interpretation for our results.

At line 446, what are the behavioral responses?

→ Behavioral responses refer to changes in the call rate. We have added this to the text.

P13 L492-494: These results suggest that turbine noise induces changes in the endocrine and immune systems as processes resulting from behavioral responses such as change of call rate rather than direct stress.

At line 455, as we know that the concentration of corticosterone is important in the calling parameters of male frogs, why did the corticosterone level not increase with call rate in the noise exposure experiment in your study?

→ The frog calling itself is one of the behavioral characteristics that require a lot of energy mobilization, and there are reports that an increase in corticosterone is made for this [32]. However, there have been studies of this during natural reproduction, but it is not yet clear when exactly this reaction occurs [31]. Our study has shown that wind noise does not elicit a direct endocrine system (HPA axis) response and increases corticosterone through changes in call rate. The increase in corticosterone with this change in call rate does not seem to happen immediately. Although it has a high peak like testosterone during the breeding season [33, 48], it seems to be not more sensitive to certain factors. Especially in the long run, it seems that it can increase as energy investment in calling increases.

P13 L482-490: Similarly, our results showed that the calling pattern of male frogs was altered by wind turbine noise. Contrari-ly, in the exposure experiment, no change was observed in corticosterone or immunity in response to the turbine noise. In certain frog species, acute stress induces an immediate increase in corticosterone responses within 30 min to 1 h [54]. Additionally, chronic stress, such as constant exposure to traffic noise, can cause chronically high levels of corti-costerone [26]. However, in our study, exposure to turbine noise for 24 h did not cause this physiological response. We, therefore, postulate that wind turbine noise may not be the stressor that increase in corticosterone by causing the direct response of HPA axis.

P13-14 L510-523: In our study, although there was no difference in testosterone between the groups, it was confirmed that sexual responses, such as an increase in call rate, induce hormonal changes (corticosterone), leading to immunosuppression, similar to that described in the ICHH. In addition to testosterone, since there were no differences in body composition that could indicate energy storage states, we believe that the energy used for advertisement calling may have caused a trade-off in energy used for immunity. Additionally, the increase in call rate and decrease in immunity caused by an increase in corticosterone, seems to support our opinion. However, it seems unlikely that this process occurs in a short period of time, as the exposure experiments did not detect these responses. We con-ducted the field investigations during the middle to end of the breeding season. Thus, be-fore the frogs were collected, they may have continuously experienced changes in call rates caused by wind turbine noise. Since our study did not attempt to compare these short- and long-term effects, further studies are needed.

Round 2

Reviewer 4 Report

The authors’ reply did not empirically solve or properly explain the first question of my previous comments, and the revised Fig. 1(a) in the authors’ revised manuscript further highlights the problems of the sampling designs of this research. In my previous comments, I mentioned that the variables (such as using of herbicides and fertilizers, pH value, soil type, abundance and diversity of prey and predators, parasites, pathogen and disease, wind velocity, ground vibration, human activity, etc.) may cause the physiological and immunological differences between the two groups (turbine vs control) and need to be verified otherwise the conclusion of this study cannot be scientifically supported. Fig. 1(a) obviously reveals that all the investigation area of turbine closed to the edges of the wetland (in purple color) while all the investigation area of control located more far away from the wetland. That is, the sampling sites between the two groups may have many differences on the habitat quality, which will interfere your results and make you wrongly deduce the turbine effects. This is a critical defect on sampling site selection that cannot be ignored. The aforementioned interfering factors may therefore cause the inconsistent results in your study that the wind turbine group in the field had a higher corticosterone level and lower immunity than the control group, while those in the noise exposure experiment has no differences on the corticosterone level or the immunity reaction post-exposure compared to pre-exposure. In addition, neither the pseudoreplication problems (from nested levels of variables) at the 4th question of my previous comments nor the problems of small sample size in this study are empirically solved by the authors. The authors also need to sample at more time points (besides 24 h) in the noise exposure experiment to correctly verify the direct effects of noise exposure on frogs.

Author Response

Reviewer

The authors’ reply did not empirically solve or properly explain the first question of my previous comments, and the revised Fig. 1(a) in the authors’ revised manuscript further highlights the problems of the sampling designs of this research. In my previous comments, I mentioned that the variables (such as using of herbicides and fertilizers, pH value, soil type, abundance and diversity of prey and predators, parasites, pathogen and disease, wind velocity, ground vibration, human activity, etc.) may cause the physiological and immunological differences between the two groups (turbine vs control) and need to be verified otherwise the conclusion of this study cannot be scientifically supported. Fig. 1(a) obviously reveals that all the investigation area of turbine closed to the edges of the wetland (in purple color) while all the investigation area of control located more far away from the wetland. That is, the sampling sites between the two groups may have many differences on the habitat quality, which will interfere your results and make you wrongly deduce the turbine effects. This is a critical defect on sampling site selection that cannot be ignored. The aforementioned interfering factors may therefore cause the inconsistent results in your study that the wind turbine group in the field had a higher corticosterone level and lower immunity than the control group, while those in the noise exposure experiment has no differences on the corticosterone level or the immunity reaction post-exposure compared to pre-exposure. In addition, neither the pseudoreplication problems (from nested levels of variables) at the 4th question of my previous comments nor the problems of small sample size in this study are empirically solved by the authors. The authors also need to sample at more time points (besides 24 h) in the noise exposure experiment to correctly verify the direct effects of noise exposure on frogs.

1) Environment parameters

→ We agree with the reviewer's comments. Many environmental, climatic and genetic factors can influence physiological and immunological parameters. Some of these contents have already been studied, and some have not been studied. As reviewers will understand, ecologists do not have “complete” control over these environmental factors in the study of ecosystems. We tried to control these factors “as much as possible”. It is for this reason that we conducted experiments at specific times within specific field investigation areas. This content also exists in Discussion. Agricultural activity occurred in the same way and at the same time within an area. Additionally, since the range of the investigation site is not very large, we expected that there would be no significant difference in climatic factors. This was demonstrated by measuring the temperature and humidity parameters. The investigation area is a landfill site, which also occurred at the same time and under the same conditions. The reviewer stated that all investigation areas of the turbine group were in wetlands and all investigation areas of the control group were outside the wetlands. This is also a reason to select paddy fields with and without wind turbines, but this may not be meaningful as the habitat of frog does not actually cover the entire area. Instead, we tried to fit turbine and control groups within each habitat fragment. That's why we set up a turbine group and a control group at the bottom of the mountain, a turbine group and a control group in the middle piece, and a turbine group and a control group in a line at the bottom.

P14 L535-544: Additionally, we collected frogs at specific season and set up the paired sites within a relatively small area range, for our purposes to minimize other environmental factors. In particular, this was very important because the behavioral, physiological, and immunological contents that we were measuring were contents that changed within a short period of time. Nevertheless, our sample size and replication still may be to be lacking. Also, we do not consider the area itself as a measurement value, but consider each site in the area as a measurement value. However, this scope range may be small, so the statistical test may over-estimate the degrees of freedom because of the clustered nature of the data. Hence, additional studies with a wider range and larger study populations, are needed.

2) Genetic differences in populations

→ Although we did not confirm the genetic difference in this study, we determined that the population of the field investigation groups was likely to be a single population through the genetic structure of Japanese tree frogs (Dryophytes japonicus) and black-spotted pond frogs (Pelophylax nigromaculatus) confirmed in Korea. Both species are common species widely distributed in Korea. The Japanese tree frogs used in this study is a species that uses separate breeding and non-breeding habitats and moves between these two places at season, whereas the other common species, the black-spotted pond frog, is a species with strong philopatry to breeding sites and non-breeding habitats (Hirai and Matsui, 2002).

→ Previously, when the genetic structure of Japanese tree frogs distributed in East Asia including Korea peninsula was confirmed, the tree frog populations in the Korean peninsula, especially in South Korea, were all identified as populations with the same genetic structure (Dufresnes et al., 2016). However, because this study used the cytochrome-b gene to determine the population structure, it may have grouped the population more broadly than other population genetic structure analysis techniques such as microsatellite analysis. We are still trying to figure out the population structure by finding or developing microsatellite markers for tree frogs, but this has not been done yet. Instead, there is a result of confirming the genetic structure of black-spotted pond frog, for which the marker has already been developed, in the largest scale of stream distributed in South Korea (Park et al., 2021). In general, species with a high degree of philopatry exhibit higher population genetic patterns and can be more clearly distinguished (Beebee, 2005). Nevertheless, the genetic origins and structures were similar among the most river basin except for one basin (Park et al., 2021). Since this pattern was also observed in the highly philopatry black-spotted pond frog, we expect a similar pattern when analyzing the finer genetic structure of Japanese tree frogs with less philopatry. We think so, as the turbine group and control group were not far apart in the field investigation. In addition, the sites investigated from field investigation group and noise exposure group, were previously confirmed to have similar genetic structures in both Japanses tree frogs and black-spotted pond frogs (Dufresnes et al., 2016; Park et al., 2021). We therefore assume that these populations will all have the same genetic structure, but we have not included them in the main text because we have not studied them precisely.

→ Also, since the wind farm at the site we investigated were completed about 2015, we believe that the possibility of genetic adaptation in perennial frogs is low.

Beebee, T.J.C., 2005. Conservation genetics of amphibians. Heredity 95, 423-427.

Hirai, T., Matsui, M., 2002. Feeding relationships between Hyla japonica and Rana nigromaculata in rice fields of Japan. J. Herpetol. 2002, 662-667.

Dufresnes, C., Litvinchuk, S.N., Borzée, A., Jang, Y., Li, J.T., Miura, I., Perrin, N., Stöck, M., 2016. Phylogeography reveals an ancient cryptic radiation in East-Asian tree frogs (Hyla japonica group) and complex relationships between continental and island lineages. BMC Evol. Biol. 16, 1-14.

Park, J.K., Yoo, N., & Do, Y., 2021. Genetic diversity and population genetic structure of black-spotted pond frog (Pelophylax nigromaculatus) distributed in South Korean River basins. Proceedings of NIE 2, 120-128.

2) Sample size and replication

→ In order to select the most suitable samples, we selected sample size within suitable conditions from the initial sample size. For this reason, the number of samples has been reduced and we also agree that this sample number is small. Since this cannot be addressed right away, we have listed these limitations and future research in discussion.

P3 L133-135: We investigated 27 male frogs in three wind farm areas (each investigation areas; about 1.7 to 1.8 km2) with a wind power generator (2.0 MW) from Yeonggwang-gun (wind tur-bine group).

P3-4 L145-150: Frogs were considered unsuitable for analysis if the frogs escaped during the sound recording process, the frogs could not be collected after recording, the call note of the frogs was not clear due to the calling of other frogs or ambient noise during the sound analysis process, there was blood on the swab during the saliva extraction process, or the amount of blood collected from the frogs was insufficient. As a result, 15 frogs were included in analysis for the wind turbine group.

P4 L166-167: Simultaneously, 24 male frogs were observed in three paddy areas (1.7 to 1.8 km2) without wind power generators (control group) from Yeonggwang-gun.

P4 L173-175: Frogs deemed unsuitable for analysis (according to the same criteria outlined for the wind turbine group) were excluded, resulting in the selection of 15 control frogs.

P14 L540-544: Also, we do not consider the area itself as a measurement value, but consider each site in the area as a measurement value. However, this scope range may be small, so the statistical test may over-estimate the degrees of freedom because of the clustered nature of the data. Hence, additional studies with a wider range and larger study populations, are needed.

3) Analysis of responses in more time points

→ We conducted noise exposure experiments to determine whether wind turbine noise could trigger a direct endocrine response. We decided that 24 hours was enough time to find out, so we experimented. However, according to the reviewer's opinion, when sampling is performed from more time points, the amount of information that can be found is likely to increase.